# OFFLINE META-REINFORCEMENT LEARNING WITH ONLINE SELF-SUPERVISION

## ABSTRACT

Meta-reinforcement learning (RL) methods can meta-train policies that adapt to new tasks with orders of magnitude less data than standard RL, but meta-training itself is costly and time-consuming. If we can meta-train on offline data, then we can reuse the same static dataset, labeled once with rewards for different tasks, to meta-train policies that adapt to a variety of new tasks at meta-test time. Although this capability would make meta-RL a practical tool for real-world use, offline meta-RL presents additional challenges beyond online meta-RL or standard offline RL settings. Meta-RL learns an exploration strategy that collects data for adapting, and also meta-trains a policy that quickly adapts to data from a new task. Since this policy was meta-trained on a fixed, offline dataset, it might behave unpredictably when adapting to data collected by the learned exploration strategy, which differs systematically from the offline data and thus induces distributional shift. We propose a hybrid offline meta-RL algorithm, which uses offline data with rewards to meta-train an adaptive policy, and then collects additional unsupervised online data, without any reward labels to bridge this distribution shift. By not requiring reward labels for online collection, this data can be much cheaper to collect. We compare our method to prior work on offline meta-RL on simulated robot locomotion and manipulation tasks and find that using additional unsupervised online data collection leads to a dramatic improvement in the adaptive capabilities of the meta-trained policies, matching the performance of fully online meta-RL on a range of challenging domains that require generalization to new tasks.

## 1 INTRODUCTION

Reinforcement learning (RL) agents are often described as learning from reward and punishment analogously to animals: in the same way that a person might train a dog by providing treats, we might train RL agents by providing rewards. However, in reality, modern deep RL agents require so many trials to learn a task that providing rewards by hand is often impractical. Meta-reinforcement learning in principle can mitigate this, by learning to learn using a set of meta-training tasks, and then acquiring new behaviors in just a few trials at meta-test time. Current meta-RL methods are so efficient that meta-trained policies require only a handful of trajectories (Rakelly et al., 2019), which is reasonable for a human to provide by hand. However, the meta-training phase in these algorithms still requires a large number of online samples, often even more than standard RL, due to the multi-task nature of the meta-learning problem.

Offline reinforcement learning methods, which use only prior experience without active data collection, provide a potential solution to this issue, because a user must only annotate multi-task data with rewards once in the offline dataset, rather than doing so in the inner loop of RL training, and the same offline multi-task data can be reused repeatedly for many training runs. While a few recent works have proposed offline meta-RL algorithms Dorfman & Tamar (2020); Mitchell et al. (2021), we identify a specific problem when an agent trained with offline meta-RL is tested on a new task: the distributional shift between the behavior policy from the offline data and the meta-test time exploration policy means that adaptation procedures learned from offline data might not perform well on the (differently distributed) data collected by the exploration policy at meta-test time. This mismatch in training distribution occurs because offline meta-RL never trains on data generated by the meta-learned exploration policy. In practice, we find that this mismatch leads to a large degradation in performance when adapting to new tasks. Moreover, we do not want to remove this distributional

**Figure 1:** (left) In offline meta-RL, an agent uses offline data from multiple tasks $T_1, T_2, \ldots$, each with reward labels that must only be provided once. (middle) In online meta-RL, new reward supervision must be provided with every environment interaction. (right) In semi-supervised meta-RL, an agent uses an offline dataset collected once to learn to generate its own reward labels for new, online interactions. Similar to offline meta-RL, reward labels must only be provided once for the offline training, and unlike online meta-RL, the additional environment interactions require neither external reward supervision nor additional task sampling.

shift by simply adopting a conservative exploration strategy, because learning an exploration strategy enables an agent to collect better data for faster adaptation.

We propose to address this challenge by collecting additional online data *without* any reward supervision, leading to a semi-supervised offline meta-RL algorithm, as illustrated in Figure 1. Online data can be relatively cheap to collect when it does not require reward labels, but it can still make it possible to bridge the distributional shift issue. To make it feasible to use this data for meta-training, we can generate synthetic reward labels for it based on the labeled offline data.

Based on this principle, we propose semi-supervised meta actor-critic (SMAC), which uses reward-labeled offline data to bootstrap a semi-supervised meta-reinforcement learning procedure, in which an offline meta-RL agent collects additional online experience without any reward labels. SMAC uses the reward supervision from the offline dataset to learn to generate new reward functions, which it uses to autonomously annotate rewards in these rewardless interactions and meta-train on this new data. Our method contains two novel contributions: first, it is the first method to combine the efficient PEARL (Rakelly et al., 2019) amortized inference method for meta-RL with AWAC (Nair et al., 2020), an effective approach for offline RL with online finetuning. This already yields an effective new offline meta-RL procedure. Second, it is the first approach to perform offline meta-RL with self-supervised online finetuning, without ground truth rewards. We evaluate our method and prior offline meta-RL methods on a number of benchmarks (Dorfman & Tamar, 2020; Mitchell et al., 2021), as well as a challenging robot manipulation domain that requires generalization to new tasks, with just a few reward-labeled trials at meta-test time. We find that, while standard meta-RL methods perform well at adapting to training tasks, they suffer from data-distribution shifts when adapting to new tasks. In contrast, our method attains significantly better performance, on par with an online meta-RL method that receives fully labeled online interaction data.

## 2 RELATED WORKS

Many prior meta-RL algorithms assume that reward labels are provided with each episode of online interaction (Duan et al., 2016; Finn et al., 2017; Gupta et al., 2018b; Xu et al., 2018; Hausman et al., 2018; Rakelly et al., 2019; Humplik et al., 2019; Kirsch et al., 2019; Zintgraf et al., 2020; Xu et al., 2020; Zhao et al., 2020; Kamienny et al., 2020). In contrast to these prior methods, our method only requires offline prior data with rewards, and additional online interaction does not require any ground truth reward signal. Prior works have also studied other formulations that combine unlabeled and labeled trials. For example, imitation and inverse reinforcement learning methods use offline demonstrations to either learn a reward function (Abbeel & Ng, 2004; Finn et al., 2016; Ho & Ermon, 2016; Fu et al., 2017) or to directly learn a policy (Schaal, 1999; Ross & Bagnell, 2010; Ho & Ermon, 2016; Reddy et al., 2019; Peng et al., 2020). Semi-supervised and positive-unlabeled reward learning (Xu & Denil, 2019; Zolna et al., 2020; Konyushkova et al., 2020) methods use reward labels provided for some interactions to train a reward function for RL. However, all of these methods have been studied in the context of a single task. In contrast, we focus on meta-learning an RL procedure that can adapt to new reward functions. In other words, we do not focus on recovering a single reward function, because there is no single test time reward or task.

SMAC uses a context-based adaptation procedure similar to that proposed by Rakelly et al. (2019), which is related to contextual policies, such as goal-conditioned reinforcement learning (Kaelbling, 1993; Schaul et al., 2015; Andrychowicz et al., 2017; Pong et al., 2018; Colas et al., 2018; Warde-

Farley et al., 2018; Péré et al., 2018; Nair et al., 2018) or successor features (Kulkarni et al., 2016; Barreto et al., 2017; 2019; Grimm et al., 2019). In contrast, our meta-learning procedure applies to any RL problem, does not assume that the reward is defined by a single goal state or fixed basis function, and only requires reward labels for static offline data.

Our method addresses a similar problem to prior offline meta-RL methods (Mitchell et al., 2021; Dorfman & Tamar, 2020). In our comparisons, we find that after offline meta-training, our method is competitive to these prior approaches, but that with additional self-supervised online fine-tuning, our method significantly outperforms these methods by mitigating the aforementioned distributional shift issue. Our method addresses the distribution shift problem by using online interactions without reward supervision. In our experiments, we found that SMAC greatly improves performance on both training and held-out tasks. Lastly, SMAC is also related to unsupervised meta-learning methods (Gupta et al., 2018a; Jabri et al., 2019), which annotate data with their own rewards. In contrast to these methods, we assume that there exists an offline dataset with reward labels that we can use to learn to generate similar rewards.

## 3 PRELIMINARIES

**Meta-reinforcement learning.** In meta-RL, we assume there is a distribution of tasks $p_{\mathcal{T}}(\cdot)$. A task $\mathcal{T}$ is a Markov decision process (MDP), defined by a tuple $\mathcal{T} = (\mathcal{S}, \mathcal{A}, r, \gamma, p_0, p_d)$, where $\mathcal{S}$ is the state space, $\mathcal{A}$ is the action space, $r$ is a reward function, $\gamma$ is a discount factor, $p_0(\mathbf{s}_0)$ is the initial state distribution, and $p_d(\mathbf{s}_{t+1} \mid \mathbf{s}_t, \mathbf{a}_t)$ is the environment dynamics distribution. A replay buffer $\mathcal{D}$ is a set of state, action, reward, next-states tuples, $\mathcal{D} = \{\mathbf{s}_i, \mathbf{a}_i, r_i, \mathbf{s}'_i\}_{i=1}^{N_{\text{size}}}$, where all the rewards come from the same task. We will use the letter $\mathbf{h}$ to denote a mini-batch or "history" and the notation $\mathbf{h} \sim \mathcal{D}$ to denote that a mini-batch $\mathbf{h}$ is sampled from a replay buffer $\mathcal{D}$. We will use the letter $\tau$ to represent a trajectory $\tau = (\mathbf{s}_1, \mathbf{a}_1, \mathbf{s}_2, \dots)$ without reward labels.

A meta-episode consists of sampling a task $\mathcal{T} \sim p_{\mathcal{T}}(\cdot)$, collecting $T$ trajectories with a policy $\pi_\theta$ with parameters $\theta$, adapting the policy to the task between trajectories, and measuring the performance on the last trajectory. Between trajectories, the adaptation procedure transforms the history of states and actions $\mathbf{h}$ from the current meta-episode into a context $\mathbf{z} = A_\phi(\mathbf{h})$, which is then given to the policy $\pi_\theta(\mathbf{a}, \mid \mathbf{s}, \mathbf{z})$ for adaptation. The exact representation of $\pi_\theta$, $A_\phi$, and $\mathbf{z}$ depends on the specific meta-RL method used. For example, the context $\mathbf{z}$ can be weights of a neural network (Finn et al., 2017) outputted by a gradient update, hidden activations ouputted by a recurrent neural network (Duan et al., 2016), or latent variables outputted by a stochastic encoder (Rakelly et al., 2019). Using this notation, the objective in meta-RL is to learn the adaptation parameters $\phi$ and policy parameters $\theta$ to maximize performance on a meta-episode given a new task $\mathcal{T}$ sampled from $p(\mathcal{T})$.

**PEARL.** Since we require an off-policy meta-RL procedure for offline meta-training, we build on probabilistic embeddings for actor-critic RL (PEARL) (Rakelly et al., 2019), an online off-policy meta-RL algorithm. In PEARL, $\mathbf{z}$ is a vector and the adaptation procedure $A_\phi$ consists of sampling $\mathbf{z}$ from a distribution $\mathbf{z} \sim q_{\phi_e}(\mathbf{z} \mid \mathbf{h})$. The distribution $q_{\phi_e}$ is generated by an encoder with parameters $\phi_e$. This encoder is a set-based neural network that processes the tuples in $\mathbf{h} = \{\mathbf{s}_i, \mathbf{a}_i, r_i, \mathbf{s}'_i\}_{i=1}^{N_{\text{enc}}}$ in a permutation-invariant manner to produce the mean and variance of a diagonal multivariate Gaussian. The policy is a contextual policy $\pi_\theta(\mathbf{a} \mid \mathbf{s}, \mathbf{z})$ conditioned on $\mathbf{z}$ by concatenating $\mathbf{z}$ to the state $\mathbf{s}$.

The policy parameter $\theta$ is trained using soft-actor critic (Haarnoja et al., 2018) which involves learning a $Q$-function, $Q_w(\mathbf{s}, \mathbf{a}, \mathbf{z})$, with parameter $w$ that estimates the sum of future discounted rewards conditioned on the current state, action, and context. The encoder parameters are trained by back-propagating the critic loss into the encoder. The actor, critic, and encoder losses are minimized via gradient descent with mini-batches sampled from separate replay buffers for each task.

**Offline reinforcement learning.** In offline reinforcement learning, we assume that we have access to a dataset $\mathcal{D}$ collected by some behavior policy $\pi_\beta$. An RL agent must train on this fixed dataset and cannot interact with the environment. One challenge that offline RL poses is that the distribution of states and actions that an agent will see when deployed will likely be different from those seen in the offline dataset as they are generated by the agent, and a number of recent methods have tackled this distribution shift issue (Fujimoto et al., 2019b;a; Kumar et al., 2019; Wu et al., 2019; Nair et al., 2020; Levine et al., 2020). Moreover, one can combine offline RL with meta-RL by training meta-RL

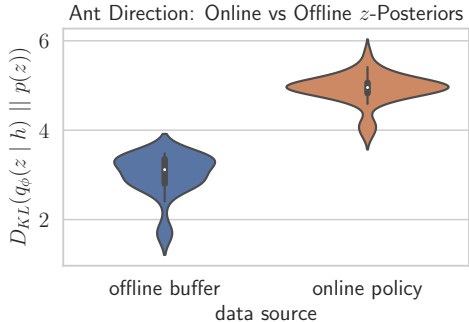 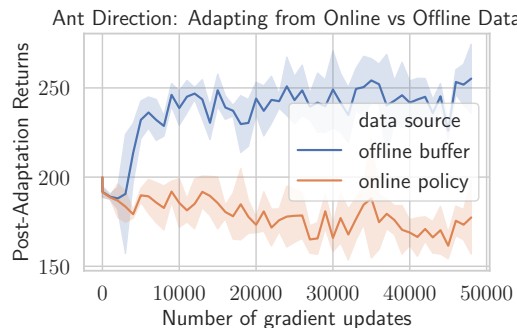

**Figure 2: Left:** The distribution of the KL-divergence between the posterior $q_{\phi_e}(\mathbf{z} \mid \mathbf{h})$ and the prior $p(\mathbf{z})$ over the course of meta-training, when conditioned on data from the offline dataset (blue) or on online data from the learned policy (orange). Data from the learned policy results in posteriors that are substantially farther from the prior, suggesting a significant difference in distribution over $\mathbf{z}$. Note that online data from the learned policy is *not* available for meta-training, but only used for measurement. **Right:** The performance of the policy after adaptation when adapted using data from the offline dataset (blue) or the learned policy (orange). Although the meta-RL policy adapts well when conditioned on $\mathbf{z}$ sampled from the offline dataset, the performance does not increase when $\mathbf{z}$ is sampled from the learned policy. Since the same policy is evaluated, the change in $\mathbf{z}$-distribution is likely the cause for the drop in performance.

on multiple datasets $\mathcal{D}_1, \ldots, \mathcal{D}_{N_{\text{buff}}}$ (Dorfman & Tamar, 2020; Mitchell et al., 2021), but in the next section we describe some limitations of this combination.

## 4 THE PROBLEM WITH NAÏVE OFFLINE META-REINFORCEMENT LEARNING

Offline meta-RL is the composition of meta-RL and offline RL: the objective is to maximize the standard meta-RL objective using only a fixed set of replay buffers, $\mathcal{D} = \{\mathcal{D}_i\}_{i=1}^{N_{\text{buff}}}$, where each buffer corresponds to data for one task. Offline meta-RL methods can in principle utilize the same constraint-based approaches that standard offline RL algorithms have used to mitigate distributional shift. However, they must also contend with an additional distribution shift challenge that is specific to the meta-RL scenario: distribution shift in $\mathbf{z}$-space.

Distribution shift in $\mathbf{z}$-space occurs because meta-learning requires learning an exploration policy $\pi_\theta$ that generates data for adaptation. However, offline meta-RL only trains the adaptation procedure $A_\phi(\mathbf{h})$ using offline data generated by a previous behavior policy, which we denote as $\pi_\beta$. After offline training, there will be a mismatch between this learned exploration policy $\pi_\theta$ and the behavior policy $\pi_\beta$, leading to a difference in the history $\mathbf{h}$ and in turn, in the context variables $\mathbf{z} = A_\phi(\mathbf{h})$. For example, in a robot manipulation setting, the offline dataset may contain smooth trajectories that were collected by a human teleoperator (Kofman et al., 2005). In contrast, the learned exploration may contain jittering due learning artifacts or the use of a stochastic policy. This jittering may not impede the robot from exploring the environment, but may result in a trajectory distribution shift that degrades the adaptation process, which only learned to adapt to smooth, human-generated trajectories in the offline dataset. More formally, if $p(\mathbf{z} \mid \mathbf{h}_{\text{offline}})$ and $p(\mathbf{z} \mid \mathbf{h}_{\text{online}})$ denote the marginal distribution of $\mathbf{z}$ given histories $\mathbf{h}$ sampled using offline and online data, respectively, the differences between $\pi_\theta$ and $\pi_\beta$ will lead to differences between $p(\mathbf{z} \mid \mathbf{h}_{\text{offline}})$ during offline training and $p(\mathbf{z} \mid \mathbf{h}_{\text{online}})$ at meta-test time.

To illustrate this difference, we compare $p(\mathbf{z} \mid \mathbf{h}_{\text{offline}})$ and $p(\mathbf{z} \mid \mathbf{h}_{\text{online}})$ on the Ant Direction task (see Section 6). We approximate these distributions by using the PEARL encoder discussed in Section 3, with $p(\mathbf{z} \mid \mathbf{h}) \approx q_{\phi_e}(\mathbf{z} \mid \mathbf{h})$, where $\mathbf{h}$ is sampled either from the offline data set or using the learned policy. We measure the KL-divergence observed at the end of offline training between the posterior $p(\mathbf{z} \mid \mathbf{h})$ and a fixed prior $p_{\mathbf{z}}(\mathbf{z})$, for different samples of $\mathbf{h}$. If the two distributions were the same, then we would expect the distribution of KL divergences to also be similar. However, Figure 2 shows that the two distributions are markedly different after the offline training phase of SMAC.

We also observe that this distribution shift negatively impacts the resulting policy. In Figure 2, we plot the performance of the learned policy when conditioned on $\mathbf{z}$ sampled from $q_{\phi_e}(\mathbf{z} \mid \mathbf{h}_{\text{offline}})$ compared to $q_{\phi_e}(\mathbf{z} \mid \mathbf{h}_{\text{online}})$. We see that the policy that uses offline data, $\mathbf{h}_{\text{offline}}$, leads to improvement, while

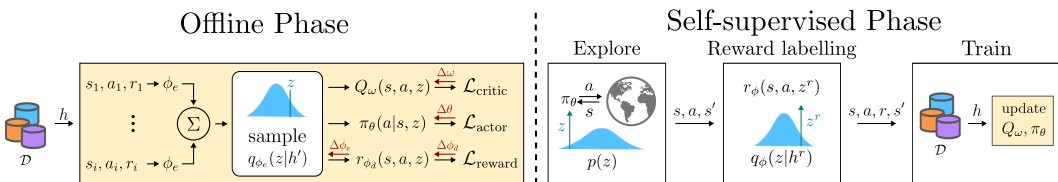

**Figure 3:** (Left) In the offline phase, we sample a history $\mathbf{h}'$ to compute the posterior $q_{\phi_e}(\mathbf{z} \mid \mathbf{h}')$. We then use a sample from this encoder and another history batch $\mathbf{h}$ to train the networks. In red, we then update the networks with $\mathbf{h}$ and the $\mathbf{z}$ sample. (Right) During the self-supervised phase, we explore by sampling $\mathbf{z} \sim p(z)$ and conditioning our policy on these observations. We label rewards using our learned reward decoder, and append the resulting data to the training data. The training procedure is equivalent to the offline phase, except that we do not train the reward decoder or encoder since no additional ground-truth rewards are observed.

the same policy that uses data from the learned policy, $\mathbf{h}_{\text{online}}$ drops in performance. Since we evaluate the same policy $\pi_\theta$ and only change how $\mathbf{z}$ is sampled, this degradation in performance suggests that the policy suffers from distributional shift between $p(\mathbf{z} \mid \mathbf{h}_{\text{offline}})$ and $p(\mathbf{z} \mid \mathbf{h}_{\text{online}})$. In other words, the encoder produces $\mathbf{z}$ vectors that are too unfamiliar to the policy when conditioned on these exploration trajectories.

We note that this issue arises in any method that trains non-Markovian policies with offline data. For example, recurrent policies for partially observed MDPs (Jaakkola et al., 1995) depend both on the current observation $\mathbf{o}$ and a history $\mathbf{h}$. When deployed, these policies must also contend with potential distributional shifts between the training and test-time history distributions, in addition to the change in observation distribution $\mathbf{o}$. This additional distribution shift may explain why many memory-based recurrent policies are often trained online (Duan et al., 2016; Heess et al., 2017; Espeholt et al., 2018) or have benefited from refreshing the memory states (Kapturowski et al., 2018). In this paper, we focus on addressing this issue specifically in the offline meta-RL setting.

**Offline meta-RL with self-supervised online training.** In complex environments where many behaviors are possible, the distribution shift in $z$-space will likely be inevitable, since the learned policy is likely to deviate from the behavior policy. To address this issue, we introduce an additional assumption: in addition to the offline dataset, we assume that the agent can autonomously interact with the environment *without observing additional reward supervision*. This problem statement is useful for scenarios where autonomously interacting with the world is relatively easy, but online reward supervision is more expensive to obtain.

Formally, we assume that the agent can generate additional rollouts in an MDP without a reward function, $\mathcal{T} \setminus r = (\mathcal{S}, \mathcal{A}, \gamma, p_0, p_d)$. These additional interactions enable the agent to explore using the learned exploration policy. Since the resulting states and actions are from the learned exploration policy, we can use them to construct $\mathbf{h}_{\text{online}}$ and enable an agent to train on the *online* context distribution, $p(\mathbf{z} \mid \mathbf{h}_{\text{online}})$, thus mitigating the distribution shift issue described above. However, meta-training requires that the history $\mathbf{h}_{\text{online}}$ contain not just states and actions, but also rewards. In the next section, we describe a method for autonomously labeling these rollouts with *synthetic* reward labels to enable an agent to meta-train on this additional data.

## 5 Semi-Supervised Meta Actor-Critic

In this section, we present our method, semi-supervised meta actor-critic (SMAC). SMAC consists of offline meta-training followed by self-supervised online meta-training to mitigate the distribution shift in $\mathbf{z}$-space. The SMAC adaptation procedure consists of passing history through the encoder described in Section 3, resulting in a posterior $q_{\phi_e}(\mathbf{z} \mid \mathbf{h})$. Below, we describe both phases.

### 5.1 Offline Meta-Training

To learn from the user-provided offline data, we adapt the PEARL (Rakelly et al., 2019) to the offline setting. Similar to PEARL, we update the critic by minimizing the Bellman error:

$$\mathcal{L}_{\text{critic}}(w) = \mathbb{E}_{(\mathbf{s},\mathbf{a},r,\mathbf{s}') \sim \mathcal{D}_i, z \sim q_{\phi_e}(\mathbf{z}|\mathbf{h}), \mathbf{a}' \sim \pi_\theta(\mathbf{a}'|\mathbf{s}',\mathbf{z})} \left[ (Q_w(\mathbf{s},\mathbf{a},\mathbf{z}) - (r + \gamma Q_{\bar{w}}(\mathbf{s}',\mathbf{a}',\mathbf{z})))^2 \right], \quad (1)$$

where $\bar{w}$ are target network weights updated using soft target updates (Lillicrap et al., 2016).

PEARL has a policy updated based on soft actor critic (SAC) (Haarnoja et al., 2018), but when naïvely applied to the offline setting, these policy updates suffer from off-policy bootstrapping error

accumulation (Kumar et al., 2019), which occurs when the target Q-function for bootstrapping $Q(\mathbf{s}', \mathbf{a}')$ is evaluated at actions $\mathbf{a}'$ outside of the training data.

To avoid error accumulation during offline training, we implicitly constrain the policy to stay close to the actions observed in the replay buffer, following the approach in a single-task offline RL algorithm called AWAC (Nair et al., 2020). AWAC uses the following loss to approximate a constrained optimization problem, where the policy is constrained to stay close to the data observed in $\mathcal{D}$:

$$\mathcal{L}_{\text{actor}}(\theta) = -\mathbb{E}_{\mathbf{s},\mathbf{a},\mathbf{s}' \sim \mathcal{D}, \mathbf{z} \sim q_{\phi_e}(\mathbf{z}|\mathbf{h})} \left[ \log \pi_\theta(\mathbf{a} \mid \mathbf{s}) \exp \left( \frac{Q(\mathbf{s},\mathbf{a},\mathbf{z}) - V(\mathbf{s}',\mathbf{z})}{\lambda} \right) \right]. \qquad (2)$$

We estimate the value function $V(\mathbf{s}, \mathbf{z}) = \mathbb{E}_{\mathbf{a} \sim \pi_\theta(\mathbf{a}|\mathbf{s},\mathbf{z})} Q(\mathbf{s}, \mathbf{a}, \mathbf{z})$ with a single sample, and $\lambda$ is the resulting Lagrange multiplier for the optimization problem. See Nair et al. (2020) for a full derivation.

With this modified actor update, we can train the encoder, actor, and critic on the offline data without the overestimation issues that afflict conventional actor-critic algorithms (Kumar et al., 2019). However, it does not address the $\mathbf{z}$-space distributional shift issue discussed in Section 4, because the exploration policy learned via this offline procedure will still deviate significantly from the behavior policy $\pi_\beta$. As discussed previously, we will aim to address this issue by collecting additional online data *without reward labels* and learning to generate reward labels if self-supervised meta-training.

**Learning to generate rewards.** To continue meta-training online without ground truth reward labels, we propose to use the offline dataset to learn a generative model over meta-training task reward functions that we can use to label the transitions collected online. Recall that during offline learning, we learn an encoder $q_{\phi_e}$ that maps experience $\mathbf{h}$ to a latent context $\mathbf{z}$ that encodes the task. In the same way that we train our policy $\pi_\theta(\mathbf{a} \mid \mathbf{s}, \mathbf{z})$ that conditionally decodes $\mathbf{z}$ into actions, we additionally train a *reward decoder* $r_{\phi_d}(\mathbf{s}, \mathbf{a}, \mathbf{z})$ with parameter $\phi_d$ [1] that conditionally decodes $\mathbf{z}$ into rewards. We train the reward decoder $r_{\phi_d}$ to reconstruct the observed reward in the offline dataset through a mean squared error loss.

Because we use the latent space $\mathbf{z}$ for reward-decoding, we back-propagate the reward decoder loss into $q_{\phi_e}$. As visualized in Figure 3, we also regularize the posteriors $q_{\phi_e}(\mathbf{z} \mid \mathbf{h})$ against a prior $p_{\mathbf{z}}(\mathbf{z})$ to provide an information bottleneck in that latent space $\mathbf{z}$ and ensure that samples from $p_{\mathbf{z}}(\mathbf{z})$ represent meaningful latent variables. We found it beneficial to not back-propagate the critic loss into the encoder, in contrast to prior work such as PEARL. To summarize, we train the encoder and reward decoder by minimizing the following loss, in which we assume that $\mathbf{z} \sim q_{\phi_e}(\mathbf{h})$:

$$\mathcal{L}_{\text{reward}}(\phi_d, \phi_e, \mathbf{h}, \mathbf{z}) = \sum_{(\mathbf{s},\mathbf{a},r) \in \mathbf{h}} \|r - r_{\phi_d}(\mathbf{s}, \mathbf{a}, \mathbf{z})\|_2^2 + D_{\text{KL}}\left( q_{\phi_e}(\cdot \mid \mathbf{h}) \big\| p_{\mathbf{z}}(\cdot) \right). \qquad (3)$$

## 5.2 Self-Supervised Online Meta-Training

We now describe the self-supervised online training procedure, during which we use the reward decoder to provide supervision. First, we collect a trajectory $\tau$ by rolling out our exploration policy $\pi_\theta$ conditioned on a context sampled from the prior $p(\mathbf{z})$. To emulate the offline meta-training supervision, we would like to label $\tau$ with rewards that are in the distribution of meta-training tasks. As such, we sample a replay buffer $\mathcal{D}_i$ uniformly from $\mathcal{D}$ to get a history $\mathbf{h}_{\text{offline}} \sim \mathcal{D}_i$ from the offline data. We then sample from the posterior $\mathbf{z} \sim q_{\phi_e}(\mathbf{z} \mid \mathbf{h}_{\text{offline}})$ and label the reward $r_{\text{generated}}$ of a new state and action, $(\mathbf{s}, \mathbf{a})$, using the reward decoder

$$r_{\text{generated}} = r_{\phi_d}(\mathbf{s}, \mathbf{a}, \mathbf{z}), \qquad \text{where } \mathbf{z} \sim q_{\phi_e}(\mathbf{z} \mid \mathbf{h}). \qquad (4)$$

We then add the labeled trajectory to the buffer and perform actor and critic updates as in offline meta-training. Lastly, since we do not observe additional ground-truth rewards, during the self-supervised phase we do not update the reward decoder $r_{\phi_d}$ or encoder $q_{\phi_e}$ and instead only train the policy and Q-function. We visualize this procedure in Figure 3.

We note that the distribution shift discussed in Section 4 is not an issue for the reward decoder. The distribution shift only occurs when we sample $\mathbf{z}$ from the encoder using online data, i.e. $\mathbf{z} \sim q_{\phi_e}(\mathbf{z} \mid \mathbf{h}_{\text{online}})$, but, we only sample from this reward decoder using $\mathbf{z}$ sampled from the encoder using *offline*

---

[1]With this notation, meta-parameters are the encoder and decoder parameters (i.e., $\phi = \{\phi_e, \phi_d\}$).

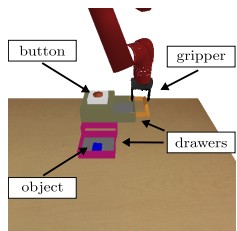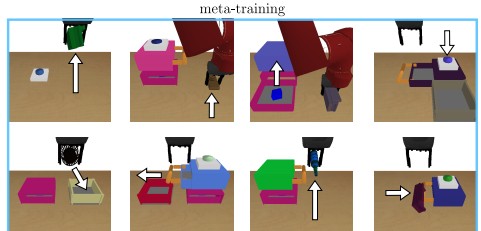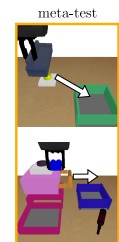

**Figure 4:** We propose a new meta-learning evaluation domain based on the environment from Khazatsky et al. (2021), in which a simulated Sawyer gripper can perform various manipulation tasks such as pushing a button, opening drawers, and picking and placing objects. We show a subset of meta-training (blue) and meta-test (orange) tasks. Each task contains a unique object configuration, and we test the agent on held-out tasks.

data, i.e. $\mathbf{z} \sim q_{\phi_e}(\mathbf{z} \mid \mathbf{h}_{\text{offline}})$. The reward decoder does need to generalize to new states and actions, and we hypothesize that reward prediction generalization is easier than policy state, action, and $\mathbf{z}$ generalization. If true, then we would expect that using the reward decoder to label rewards and train on those labels, as in SMAC, will outperform directly using the policy on new tasks (as in Figure 2).

### 5.3 ALGORITHM SUMMARY AND DETAILS

We visualize SMAC in Figure 3. For offline training, we assume access to offline datasets $\mathcal{D} = \{\mathcal{D}_i\}_{i=1}^{N_{\text{buff}}}$, where each buffer corresponds to data generated for one task. Each iteration, we sample a buffer $\mathcal{D}_i \sim \mathcal{D}$ and a history from this buffer $\mathbf{h} \sim \mathcal{D}_i$. We condition the stochastic encoder $q_{\phi_e}$ on this history to obtain a sample $\mathbf{z} \sim q_{\phi_e}(\mathbf{z} \mid \mathbf{h})$. We then use this sample $\mathbf{z}$ and a second history sample $\mathbf{h}' \sim \mathcal{D}_i$ to update the $Q$-function, the policy, encoder, and decoder by minimizing Equation (1), Equation (2), and Equation (3) respectively. During the self-supervised phase, we found it beneficial to train the actor with a combination of the loss in Equation (2) and the original PEARL actor loss, weighted by hyperparameter $\lambda_{\text{pearl}}$. We provide pseudo-code for SMAC in Appendix A.

## 6 EXPERIMENTS

We presented a method that uses self-supervised data to mitigate the $\mathbf{z}$-space distribution shift that occurs in offline meta-RL. In this section, we evaluate how well the self-supervised phase of SMAC mitigates the resulting drop in performance, and we compare SMAC to prior offline meta-RL methods on a range of meta-RL benchmark tasks that require generalization to unseen tasks at meta-test time.

**Meta-RL tasks.** We first evaluate our method on multiple simulated MuJoCo (Todorov et al., 2012) meta-learning tasks that have been used in past online and offline meta-RL papers (Finn et al., 2017; Rakelly et al., 2019; Dorfman & Tamar, 2020; Mitchell et al., 2021) (see Figure 9). Although there are standardized benchmarks for offline RL (Fu et al., 2020) and meta-RL (Yu et al., 2020), there are no standard benchmarks for offline meta-RL particular, so we use a combination of tasks in prior work (Dorfman & Tamar, 2020; Mitchell et al., 2021), and introduce a more complex robotic manipulation domain meant to push task generalization to the limit.

We first evaluate five different domains, Cheetah Velocity, Ant Direction, Humanoid, Walker Param, and Hopper Param, which have been used in prior work (Rothfuss et al., 2018; Rakelly et al., 2019; Dorfman & Tamar, 2020). In the first three domains, the agent is rewarded for moving in an unobserved target velocity and a meta-episode consists of sampling a desired velocity. The last task, Humanoid, is particularly challenging due to the high space dimension of 376. The last two domains, Walker Param and Hopper Param, require adapting to different dynamics parameters, such as friction, joint mass, and inertia. The agent must adapt to each task within $T = 3$ episodes, each of length 200.

We also evaluated SMAC on a significantly more diverse robot manipulation meta-learning task called Sawyer Manipulation, based on the goal-conditioned environment introduced by Khazatsky et al. (2021). This is a simulated PyBullet environment Coumans & Bai (2016–2021) in which a Sawyer robot arm can manipulate various objects. Sampling a task $\mathcal{T} \sim p(\mathcal{T})$ involves sampling both a new configuration of the environment and the desired behavior to achieve, such as pushing a button, opening a drawer, or lifting an object (see Figure 4). The sparse reward is 0 when the desired behavior is achieved and $-1$ otherwise. The action space consists of 3 dimensions to control the end-effector in Euclidean space and one dimension to control the gripper. The state is a 13-dimensional vector,

which we detail in Appendix C. The sparse reward, precise manipulation requirement, and diversity of object configuration between each task sample make this environment difficult.

In all of the environments, we test the meta-RL procedure's ability to generalize to new tasks by evaluating the policies on *held-out tasks* sampled from the same distribution as in the offline datasets. We give a complete description of environments and task distribution in Appendix C.

**Offline data collection.** For the MuJoCo tasks, we use the replay buffer from a single PEARL run that uses the ground-truth reward. We limit the data collection to 1200 transitions (i.e., 6 trajectories) per task and terminate the PEARL run early, forcing the meta-RL agent to learn from highly suboptimal data. For Sawyer Manipulation, we collect data using a scripted policy that randomly performs one of many possible tasks in the environment. We used 50 training tasks and 50 trajectories of length 75 each. Note that the performed task (e.g. open drawer) may differ from the task that is rewarded (e.g. push a button). As a result, in the offline dataset, the robot succeeds on the task in 46% of the transitions, indicating that this data is highly suboptimal. In contrast to prior work (Dorfman & Tamar, 2020; Mitchell et al., 2021), we use this limited data because it enables us to test how well the different methods can improve over suboptimal, offline data. See Appendix C for further discussion and details.

**Comparisons and ablations.** As an upper bound, we include the performance of PEARL with online training using oracle ground-truth rewards rather than self-generated rewards, which we label *Online Oracle*. To understand the importance of using the actor loss in Equation (2), we include an ablation that replaces Equation (2) with the actor loss from PEARL, which we label *SMAC (actor ablation)*. We also include a meta-imitation baseline, labeled *meta behavior cloning*, which replaces the actor update in Equation (2) with simply maximizing $\log \pi_\theta(\mathbf{a} \mid \mathbf{s}, \mathbf{z})$. A gap between SMAC and this imitation method would help us understand whether or not our method improves over the (possibly sub-optimal) behavior policy. For comparisons to prior work, we include the two previously proposed offline meta-RL methods: meta-actor critic with advantage weighting (labelled *MACAW*) (Mitchell et al., 2021) and Bayesian offline RL (labelled *BORеL*) (Dorfman & Tamar, 2020). MACAW and BOReL assume that rewards are provided during training and cannot make use of the self-supervised online interaction, and so we cannot compare directly to these past works. Therefore, we report their performance only after offline training. For these prior work, we used the open-sourced code directly from each paper (PEARL, MACAW, and BOReL), and and trained these methods using the same offline dataset. To ensure a fair comparison, we ran the original hyperparameters and matched SMAC hyperparameters (matching network size, learning rate, and batch size), taking the better of the two as the result for each prior method. We include a list of all hyperparameters and network architectures in Appendix C.3.

**Comparison results.** We plot the mean post-adaptation returns and standard deviation across 4 seeds in Figure 5. We see that across all three environments, SMAC consistently improves during the self-supervised phase, and often achieves a similar performance to the oracle that uses ground-truth reward during the online phase of learning. SMAC also significantly improves over meta behavior cloning, which confirms that the data in the offline dataset is far from optimal.

We also observed even *before* the online phase, our method is competitive with BOReL and MACAW as a stand-alone offline meta-RL method. When self-supervised training is included, we see that SMAC significantly improves over the performance of BOReL on all three tasks and significantly improves over MACAW on two of the three tasks, including the Sawyer Manipulation environment, which is by far the most challenging and exhibits the most variability between tasks. In this domain, we also see the largest gains from the AWAC actor update, in contrast to the actor ablation in blue, which corresponds to a PEARL-style update, indicating that properly handling the offline phase is also important for good performance.

In conclusion, our method is the only one attains generalization performance close to the Online Oracle upper bound baseline on all three tasks, indicating that self-supervised online fine-tuning is highly effective for mitigating distributional shift in meta-RL, whereas without it offline meta-RL generally does not exceed the performance of a meta-imitation learning baseline.

**Visualizing the distribution shift.** Lastly, we further investigate if the self-supervised training helps specifically because it mitigates a distribution shift caused by the exploration policy. To investigate this, we visualize the trajectories of the learned policy both before and after the self-supervised phase for the Ant Direction task in Figure 6. For each plot, we show trajectories from the

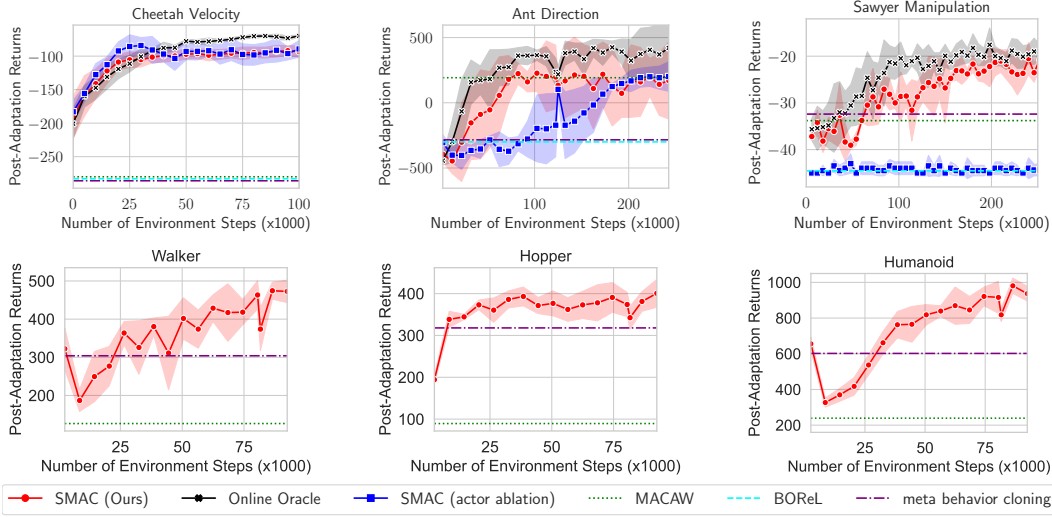

**Figure 5:** We report the final return of meta-test adaptation on unseen test tasks versus the amount of self-supervised meta-training following offline meta-training. Our method SMAC, shown in red, consistently trains to a reasonable performance from offline meta-RL (shown at step 0) and then steadily improves with online self-supervised experience. The offline meta-RL methods, MACAW Mitchell et al. (2021) and BOReL are competitive with the offline performance of SMAC but have no mechanism to improve via self-supervision. We also compare to SMAC (SAC ablation) which uses SAC instead of AWAC as the underlying RL algorithm. This ablation struggles to train a value function offline, and so struggles to improve on more difficult tasks.

policy $\pi_\theta(\mathbf{a} \mid \mathbf{s}, \mathbf{z})$ when the encoder $q_{\phi_e}(\mathbf{z} \mid \mathbf{h})$ is conditioned on histories from either the offline dataset ($\mathbf{h}_{\text{offline}}$) or from the learned exploration policy ($\mathbf{h}_{\text{online}}$). Since the same policy is evaluated, differences between the resulting trajectories represent the distribution shift caused by using history from the learned exploration policy rather than from the offline dataset.

We see that before the self-supervised phase, there is a large difference between the two modes that can only be attributed to the difference in $\mathbf{h}$. When using $\mathbf{h}_{\text{online}}$, the post-adaptation policy only explores one mode, but when using $\mathbf{h}_{\text{offline}}$, the policy moves in all directions. This qualitative difference explains the large performance gap observed in Figure 2 and highlights that the adaptation procedure is sensitive to the history $\mathbf{h}$ used to adapt. In contrast, after the self-supervised phase, the policy moves in all directions regardless of where the history came from. In Appendix B, we also visualize the exploration trajectories and found that the exploration trajectories are qualitatively similar both before and after the self-supervised phase. Together, these results illustrate the SMAC policy learns to adapt to the exploration trajectories by using the self-supervised phase to mitigate the distribution shift that occurs with naïve offline meta RL.

# 7 CONCLUSION

In this paper, we studied a problem specific to offline meta-RL: distribution shift in the context parameters $\mathbf{z}$. This distribution shift occurs because the data collected by the meta-learned exploration policy differs from the offline dataset. To address this problem, we assumed that an agent can sample new trajectories without additional reward labels and presented SMAC, a method that uses these additional interactions together with state-of-the-art offline RL techniques to provide an effective meta-RL method. Experimentally, we found that SMAC significantly improves over the performance of prior offline meta-RL methods, after the self-supervised online phase.

A limitation of SMAC is that it assumes that the agent can gather additional unlabeled online samples. However, this assumption may not be satisfied due to safety concerns and engineering needed to ensure that automatic resets are feasible, and therefore enabling safe, autonomous environment interactions is an important direction for future research.

Lastly, using self-supervised interaction to mitigate distribution shift in offline meta RL is orthogonal to the underlying meta RL algorithm. In this paper, we built off of PEARL (Rakelly et al., 2019) due to its simplicity, but this insight could easily be used to improve other existing (Mitchell et al., 2021; Dorfman & Tamar, 2020) or future offline meta RL algorithms.

## REPRODUCIBILITY STATEMENT

To ensure reproducibility, we have included the source code for our method and environments in the supplementary file, which includes a README file that describes how to install our code and run the experiments. We describe in detail the environments and hyperparameters used for the experiment in Appendix C. For clarity, we have also included pseudo-code of SMAC on the first page of the supplementary material.

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

# Supplementary Material

## A  METHOD PSEUDO-CODE

We present the pseudo-code for SMAC in Algorithm 1.

---

**Algorithm 1** Semi-Supervised Meta Actor-Critic

---

1:  Input: datasets $\mathcal{D} = \{\mathcal{D}_i\}_{i=1}^{N_{\text{buff}}}$, policy $\pi_\theta$, Q-function $Q_w$, encoder $q_{\phi_e}$, and decoder $r_{\phi_d}$.
2:  **for** iteration $n = 1, 2, \ldots, N_{\text{offline}}$ **do**                                    ▷ offline phase
3:      Sample buffer $\mathcal{D}_i \sim \mathcal{D}$ and two histories from buffer $\mathbf{h}, \mathbf{h}' \sim \mathcal{D}_i$.
4:      Use the first history sample to $\mathbf{h}$ to infer $\mathbf{z}$ encode it $\mathbf{z} \sim q_{\phi_e}(\mathbf{h})$.
5:      Update $\pi_\theta, Q_w, q_{\phi_e}, r_{\phi_d}$ by minimizing $\mathcal{L}_{\text{actor}}, \mathcal{L}_{\text{critic}}, \mathcal{L}_{\text{reward}}$ with samples $\mathbf{z}, \mathbf{h}'$.
6:  **for** iteration $n = 1, 2, \ldots, N_{\text{online}}$ **do**                                    ▷ self-supervised phase
7:      Collect trajectory $\tau$ with $\pi_\theta(\mathbf{a} \mid \mathbf{s}, \mathbf{z})$, with $\mathbf{z}_t \sim p(\mathbf{z})$.
8:      Sample buffer $\mathcal{D}_i \sim \mathcal{D}$ and offline history $\mathbf{h}_{\text{offline}} \sim \mathcal{D}_i$.
9:      Use $\mathbf{h}_{\text{offline}}$ to label the rewards in $\tau$, as in Equation (4), and add the resulting data to $\mathcal{D}_i$.
10:     Sample buffer $\mathcal{D}_i \sim \mathcal{D}$ and two histories from buffer $\mathbf{h}, \mathbf{h}' \sim \mathcal{D}_i$.
11:     Encode first history $\mathbf{z} = q_{\phi_e}(\mathbf{h})$.
12:     Update $\pi_\theta, Q_w$ by minimizing $\mathcal{L}_{\text{actor}}, \mathcal{L}_{\text{critic}}$ with samples $\mathbf{z}, \mathbf{h}'$.

---

## B  ADDITIONAL EXPERIMENTAL RESULTS

**Exploration and offline dataset visualization**    In Figure 7, we visualize the post-adaption trajectories generated when conditioning the encoder the online exploration trajectories $\mathbf{h}_{\text{online}}$ and the offline trajectories $\mathbf{h}_{\text{offline}}$. Similar to Figure 6, and also visualize the online and offline trajectories themselves. We see that the exploration trajectories $\mathbf{h}_{\text{online}}$ and the offline trajectories $\mathbf{h}_{\text{offline}}$ are very different (green vs red, respectively), but the self-supervised phase mitigates the negative impact that this distribution shift has on offline meta RL. In particular, the post-adaptation trajectories conditioned on these two data sources (blue and orange) are similar after the self-supervised training, whereas before the self-supervised training, only the trajectories conditioned on the offline data (blue) move in multiple directions.

**Addressing state-space distribution shift by self-supervised meta-training on test tasks.**    Another source of distribution shift that can negatively impact a meta-policy is a distribution shift in state space. While this distribution shift occurs in standard offline RL, we expect this issue to be

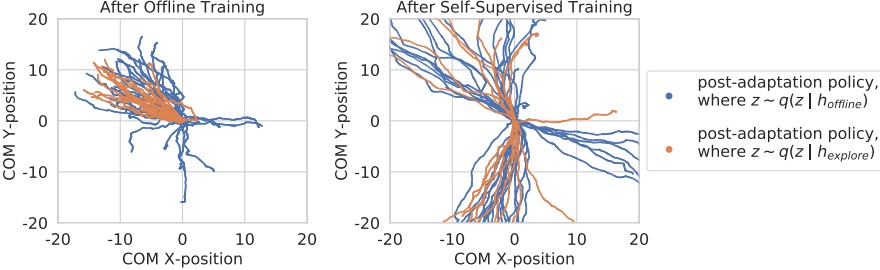

**Figure 6:** Example XY-coordinates visited by a learned policy on the Ant Direction task. **Left:** Immediately after offline training, the post adaptation policy moves in many different directions when conditioned on $\mathbf{h}_{\text{offline}}$ (blue). However, when conditioned on $\mathbf{h}_{\text{online}}$ (orange), the policy only moves up and to the left, suggesting that the post-adaptation policy is sensitive to data distribution used to collect $\mathbf{h}$. **Right:** After the self-supervised phase, the post-adaptation policy moves in many directions regardless of the data source, suggesting that the self-supervised phase mitigates the distribution shift between conditioning on offline and online data.

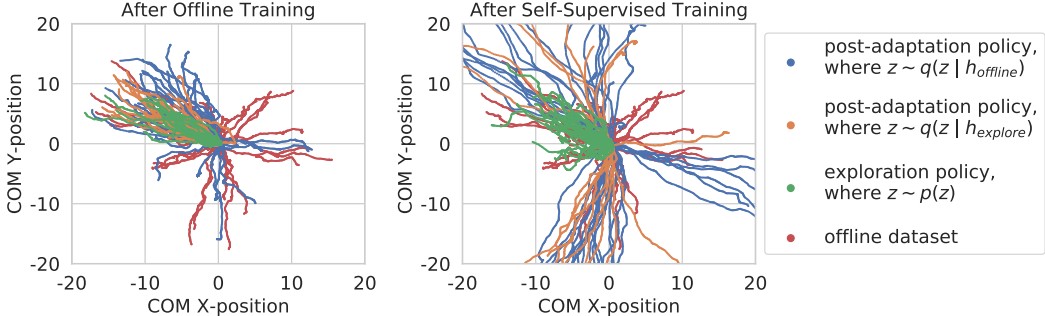

**Figure 7:** We duplicate Figure 6 but include the exploration trajectories (green) and example trajectories from the offline dataset (red). We see that the exploration policy both before and after self-supervised training primarily moves up and to the left, whereas the offline data moves in all direction. Before the self-supervised phase, we see that conditioning the encoder on online data (orange) rather than offline data (blue) results in very different policies, with the online data resulting in the post-adaptation policy only moving up and to the left. However, the self-supervised phase of SMAC mitigates the impact of this distribution shift and results in qualitatively similar post-adaptation trajectories, despite the large difference between the exploration trajectories and offline dataset trajectories.

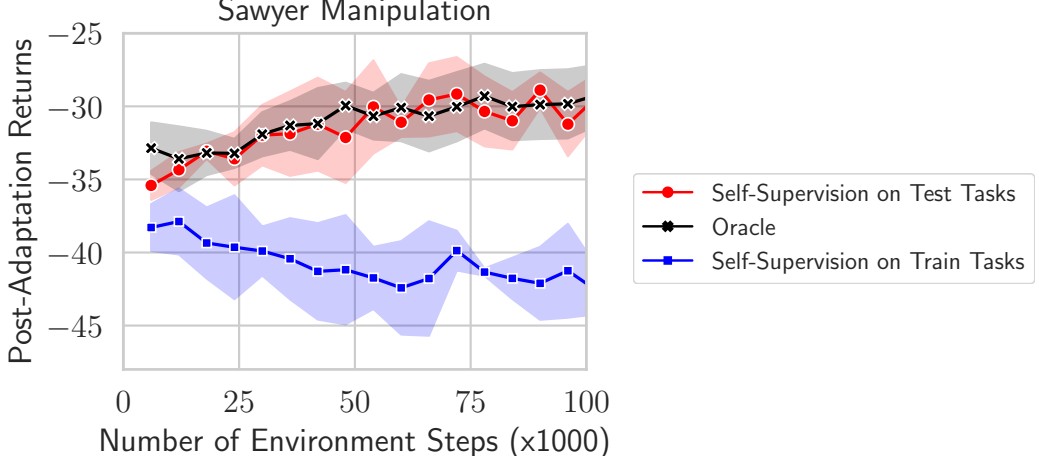

**Figure 8:** Learning curves when performing self-supervised training on the test environments (red) or the meta-training environments (blue). We also compare to an oracle that trains on test environments in combination with ground-truth rewards (black). We see that interacting with the test environment without rewards allows for steady improvement in post-adaptation test performance and obtains a similar performance to meta-training on those environments with ground-truth rewards.

more prominent in meta RL, where there is a focus on generalizing to completely novel tasks. In many real-world scenarios, experiencing the state distribution of a novel task is possible, but it is the supervision (ie. reward signal) that is expensive to obtain. Can we mitigate state distribution shift by allow the agent to meta-train in the test task environments, but without rewards?

In this experiment, we evaluate our method, SMAC, when training online on the test tasks instead of on the meta-training tasks as in the experiments in Section 6. Prior work has explored this idea of self-supervision with test tasks in supervised learning Sun et al. (2020) and goal-conditioned RL Khazatsky et al. (2021). We use the Sawyer Manipulation environment to study how self-supervised training can mitigate state distribution shifts, as these environments contain significant variation between tasks. To further increase the complexity of the environment, we use a version of the environment which samples from a set of eight potential desired behaviors instead of three.

We compare self-supervised training on test tasks to self-supervised training on the set of meta-training tasks, which are also the tasks contained in the offline dataset. A large gap in performance indicates that interacting with the test tasks can mitigate the resulting distribution shift even when no reward labels are provided.

We show the results in Figure 8 and find that there is indeed a large performance gap between the two training modes, with self-supervision on test tasks improving post-adaptation returns while self-supervision on meta-training tasks does not improve post-adaptation returns. We also compare to an oracle method that performs online training with the test tasks and the ground-truth reward signal. We see that SMAC is competitive with the oracle, demonstrating that we do not need access to rewards in order to improve on test tasks. Instead, the entire performance gain comes from experiencing the new state distribution of test tasks. Overall, these results suggest that SMAC is effective for mitigate distribution shifts in both z-space and state space, even when an agent can interact in the environment without reward supervision.

**Reward Accuracy Dependence**    Our method involves training a reward decoder, and so a natural question is: How good must the reward decoder be for SMAC to work? We found that the reward loss does not need to be particularly low for our method to work. Specifically, in the Sawyer Manipulation environment, the reward scale is either $-1$ or $0$, meaning that the maximum reward scale is $1$. We observed that the reward decoder loss is around $0.2$ on the training task and around $0.25$ on the test tasks, indicating that the method does not need a relatively low reward decoder loss to perform well.

## C    EXPERIMENTAL DETAILS

### C.1    DATA COLLECTION DIFFERENCE FROM PRIOR WORK

BOReL and MACAW were both developed assuming several orders of magnitude more data than the regime that we tested. For example, in the BOReL paper (Dorfman & Tamar, 2020), the Cheetah Velocity was trained with an offline dataset using 400 million transitions and performs additional reward relabeling using ground-truth information about the transitions. In contrast, our offline dataset contains only 240 thousand transitions, roughly *three orders of magnitude* fewer transitions. Similarly, MACAW uses 100M transitions for Cheetah Velocity, over 40 times more transitions than used in our experiments. These prior methods also collect offline datasets by training *task-specific policies*, which converge to near-optimal policies within the first million time step (Haarnoja et al., 2018), meaning that they utilize very high-quality data. In contrast, our experiments focused on the scenario with many fewer and much lower quality trajectories, which is likely the cause of the relatively worse performance of BOReL and MACAW than in the original papers.

### C.2    ENVIRONMENT DETAILS

In this section, we describe the state and action space of each environment. We also describe how reward functions were generated and how the offline data was generated.

**Ant Direction**    The Ant Direction task consists of controlling a quadruped "ant" robot that can move in a plane. Following prior work (Rakelly et al., 2019; Dorfman & Tamar, 2020), the reward function is the dot product between the agent's velocity and a direction uniformly sampled from the unit circle. The state space is $\mathbb{R}^{20}$, comprising the orientation of the ant (in quaternion) as well as the angle and angular velocity of all 8 joints. The action space is $[-1, 1]^8$, with each dimension corresponding to the torque applied to a respective joint. The reward function is the negative absolute difference between the agent's x-velocity and a target velocity uniformly sampled from $[0, 3]$.

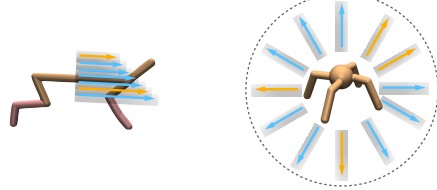

**Figure 9:** Illustrations of two evaluation domains, each of which has a set of meta-train tasks (examples shown in blue) and held out test tasks (orange). The domains include (left) a half cheetah tasked with running at different speeds and (right) a quadruped ant locomoting to different points on a circle.

The offline data is collected by running PEARL (Rakelly et al., 2019) on this meta RL task with 100 pre-sampled [2] target velocities. We terminate PEARL after 100 iterations, with each iteration

---

[2]To mitigate variance coming from this sampling procedure, we use the same sampled target velocities across all experiments and comparisons. We similarly use a pre-sampled set of tasks for the other environments.

consisting of collecting trajectories until at least 1000 new transitions have been observed. As discussed in Appendix C.1, this results in highly suboptimal data, enabling us to test how well the different methods can improve over the offline data. In PEARL, there are two replay buffers saved for each task, one for sampling data for training the encoder and another for training the policy and Q-function. We will call the former replay buffer the encoder replay buffer and the latter the RL replay buffer. The encoder replay buffer contains data generated by only the exploration policy, in which $\mathbf{z} \sim p_{\mathbf{z}}(\mathbf{z})$. The RL replay buffer contains all data generated, including both exploration and post-adaptation, in which $\mathbf{z} \sim q_{\phi_e}(\mathbf{z} \mid \mathbf{h})$. To make the offline dataset, we load the last 1200 samples of the RL replay buffer and the last 400 transitions from the encoder replay buffer into corresponding RL and encoder replay buffers for SMAC.

**Cheetah Velocity** The Cheetah Velocity task consists of controlling a two-legged "half cheetah" that can move forwards or backwards along the x-axis. Following prior work (Rakelly et al., 2019; Dorfman & Tamar, 2020), the reward function is the absolute difference product between the agent's x-velocity and a velocity uniformly sampled from $[0, 3]$. The state space is $\mathbb{R}^{20}$, comprising the $z$-position; the cheetah's x- and z- velocity; the angle and angular velocity of each joint and the half-cheetah's y-angle; and the XYZ position of the center of mass. The action space is $[-1, 1]^6$, with each dimension corresponding to the torque applied to a respective joint.

The offline data is collected in the same way as in the Ant Direction task, using a run from PEARL with 100 pre-sampled target velocities. For the offline dataset, we use the first 1200 samples from the RL replay buffer and last 400 samples from the encoder replay buffer after 50 PEARL iterations, with each iteration containing at least 1000 new transitions. For only this environment, we found that it was beneficial to freeze the encoder buffer during the self-supervised phase.

**Sawyer Manipulation** The state space, action space, and reward is described in Section 6. Tasks are generated by sampling the initial configuration, and then the desired behavior. There are five objects: a drawer opened by handle, a drawer opened by button, a button, a tray, and a graspable object. The state is a 13-dimensional vector comprising of the 3D position of the end-effector and positions of objects in the scene: 3 dimensions for the graspable object, and 1 dimension for each articulated joint in the scene including the robot grippers. If an object is not present, it takes on position 0 in the corresponding element of the state space. First, the presence or absence of each of the five is randomized. Next, the position of the drawers (from 2 sides), initial position of the tray (from 4 positions), and the object (from 4 positions) is randomized. Finally, the desired behavior is randomly chosen from the following list, but only including the ones that are possible in the scene: "move hand", "open top drawer with handle", or "open bottom drawer with button". The offline data is collected using a scripted controller that does not know the desired behavior and randomly performs potential tasks in the scene, choosing another task if it finishes one task before the trajectory ends. This data is loaded into a single replay buffer used for both the encoder and RL.

**Walker Param** This environment involves controlling a bi-pedal robot that can move along the Z- and Y-axis. The source code is taken from the `rand_param_envs` repository [3], which has been used in prior work (Rakelly et al., 2019). A task involves randomly sampling the body mass, the joint damping coefficients, the body inertia, and the friction parameters from a log-uniform distribution on the range $[1.5^{-3}, 1.5^3]$ . The reward is the velocity plus a bonus of 1 for staying alive and a control penalty of $10^{-3} \times \|a\|^2$, where $\|a\|$ is the $\ell_2$ norm of the action. The state space is 17 dimensions and the action space is 6 dimensions (one for each joint).

**Hopper Param** This environment involves controlling a one-legged robot that can move along the Z- and Y-axis. This environment is also taken from the `rand_param_envs` repository. The reward is the same as in Walker Param, and tasks are sampled the same as for the Walker Param. The state space is 11 dimensions and the action space is 3 dimensions (one for each joint).

**Humanoid** This environment is based on the Humanoid environment from OpenAI Gym (Brockman et al., 2016). We reuse the standard reward function but replace the forward-velocity reward with a velocity reward based on target direction. Each task consists of sampling a target direction

---

[3]`https://github.com/dennisl88/rand_param_envs`

| Hyperparameter | Value |
|---|---|
| RL batch size | 256 |
| encoder batch size | 64 |
| meta batch size | 4 |
| Q-network hidden sizes | $[300, 300, 300]$ |
| policy network hidden sizes | $[300, 300, 300]$ |
| decoder network hidden sizes | $[64, 64]$ |
| encoder network hidden sizes | $[200, 200, 200]$ |
| $\mathbf{z}$ dimensionality ($d_z$) | 5 |
| hidden activation (all networks) | ReLU |
| Q-network, encoder, and decoder output activation | identity |
| policy output activation | tanh |
| discount factor $\gamma$ | 0.99 |
| target network soft target $\eta$ | 0.005 |
| policy, Q-network, encoder, and decoder learning rate | $3 \times 10^{-4}$ |
| policy, Q-network, encoder, and decoder optimizer | Adam |
| # of gradient steps per environment transition | 4 |

**Table 1:** SMAC Hyperparameters for Self-Supervised Phase

| Hyperparameter | Cheetah | Ant | Sawyer | Walker | Hopper | Humanoid |
|---|---|---|---|---|---|---|
| horizon (max # of transitions per trajectory) | 200 | 200 | 50 | 200 | 200 | 200 |
| AWR $\beta$ | 100 | 100 | 0.3 | 100 | 100 | 100 |
| reward scale | 5 | 5 | 1 | 5 | 5 | 5 |
| # of training tasks | 100 | 100 | 50 | 50 | 50 | 50 |
| # of test tasks | 30 | 20 | 10 | 5 | 5 | 5 |
| # of transitions per training task in offline dataset | 1600 | 1600 | 3750 | 1200 | 1200 | 1200 |
| $\lambda_{\text{pearl}}$ | 1 | 1 | 0 | 1 | 1 | 1 |

**Table 2:** Environment Specific SMAC Hyperparameters

uniformly at random. The state space is 376 dimensions and the action space is 17 dimensions (one for each joint).

## C.3 HYPERPARAMETERS

We list the hyperparameters for training the policy, encoder, decoder, and Q-network in Table 1. If hyperparameters were different across environments, they are listed in Table 2. For pretraining, we use the same hyperparameters and train for 50000 gradient steps. Below, we give details on non-standard hyperparameters and architectures.

**Batch sizes.** The RL batch size is the batch size per task when sampling $(\mathbf{s}, \mathbf{a}, r, \mathbf{s}')$ tuples to update the policy and Q-network. The encoder batch size is the size of the history $\mathbf{h}$ per task used to conditioned the encoder $q_{\phi_e}(\mathbf{z} \mid \mathbf{h})$. The meta batch size is how many tasks batches were sampled and concatenated for both the RL and encoder batches. In other words, for each gradient update, the policy and Q-network observe (RL batch size) $\times$ (meta batch size) transitions and the encoder observes (RL batch size) $\times$ (encoder batch size) transitions.

**Encoder architecture.** The encoder uses the same architecture as in Rakelly et al. (2019). The posterior is given as the product of independent factors

$$q_{\phi_e}(\mathbf{z} \mid \mathbf{h}) \propto \prod_{\mathbf{s},\mathbf{a},r \in \mathbf{h}} \Phi(\mathbf{z} \mid \mathbf{s}, \mathbf{a}, r),$$

where each factor is a multi-variate Gaussian over $\mathbb{R}^{d_z}$ with learned mean and diagonal variance. In other words,

$$\Phi_{\phi_e}(\mathbf{z} \mid \mathbf{s}, \mathbf{a}, r) = \mathcal{N}(\mu_{\phi_e}(\mathbf{s}, \mathbf{a}, r), \sigma_{\phi_e}(\mathbf{s}, \mathbf{a}, r)).$$

The mean and standard deviation is the output of a single MLP network with output dimensionality $2 \times d_z$. The output of the MLP network is split into two halves. The first half is the mean and the second half is passed through the softplus activation to get the standard deviation.

**Self-supervised actor update.** The parameter $\lambda_{\text{pearl}}$ controls the actor loss during the self-supervised phase, which is

$$\mathcal{L}_{\text{actor}}^{\text{self-supervised}}(\theta) = \mathcal{L}_{\text{actor}}(\theta) + \lambda_{\text{pearl}} \cdot \mathcal{L}_{\text{actor}}^{\text{PEARL}}(\theta),$$

where $\mathcal{L}_{\text{actor}}^{\text{PEARL}}$ is the actor loss from PEARL (Rakelly et al., 2019). For reference, the PEARL actor loss is

$$\mathcal{L}_{\text{actor}}^{\text{PEARL}}(\theta) = \mathbb{E}_{\mathbf{s} \sim \mathcal{D}_i, \mathbf{z} \sim q_{\phi_e}(\mathbf{z}|\mathbf{h})} \left[ D_{\text{KL}} \left( \pi_\theta(\mathbf{a} \mid \mathbf{s}, \mathbf{z}) \middle\| \frac{\exp Q_w(\mathbf{s}, \mathbf{a}, \mathbf{z})}{Z(\mathbf{s})} \right) \right].$$

When the parameter $\lambda_{\text{pearl}}$ is zero, the actor update is equivalent to the actor update in AWAC (Nair et al., 2020).

**Comparisons** As discussed in Section 6, we used the authors' code for PEARL (Rakelly et al., 2019),[4] BOReL (Dorfman & Tamar, 2020),[5] and MACAW (Mitchell et al., 2021).[6] To ensure a fair comparison, we ran the original hyperparameters and matched SMAC hyperparameters (matching network size, learning rate, and batch size), taking the better of the two as the result for each prior method. For all comparisons, we evaluated the final policy using the same evaluation method as SMAC, i.e. collecting exploration trajectories with the learned policy and perform adaptation using this newly collected data.

---

[4]https://github.com/katerakelly/oyster
[5]https://github.com/Rondorf/BOReL
[6]https://github.com/eric-mitchell/macaw

