# OpenReview forum: "Offline Meta-Reinforcement Learning with Online Self-Supervision"
_ICLR.cc/2022/Conference — ICLR 2022 Submitted_

### Official Review · Reviewer_tmzu · 2021-11-01

**Correctness:** 4
**Technical Novelty And Significance:** 2
**Empirical Novelty And Significance:** 3
**Recommendation:** 5
**Confidence:** 4

**Main Review:**

Strengths:

This paper notes a problem called "distribution shift in z space" in offline meta RL and illustrates this issue in Figure 2.
This paper is well written and easy to understand.
The visualization of the distribution shift at the end of the experiment part is informative.

Weaknesses:

The problem of "distribution shift in z space" might require deeper discussion. In other context-based meta RL algorithms (e.g. RL^2, MQL, CaDM), where the task context feature is extracted from a sequence of past historical transitions, is there still the problem of  "distribution shift in z space". If yes, could the proposed unsupervised online training be beneficial to solve this issue of "distribution shift in z space"  in these algorithms?

In the online training process, is the distribution of tasks the same as tasks in the offline training dataset? Figure 1(right) seems a bit confusing. The task set T1, T2, T3, T4, T5, T6 in online training is different from the task set T1, T2, T3 in the offline set. Then when we compare with offline meta-RL (Figure 1 left) or online meta-RL (Figure 1 middle), the proposed method is exposed to more tasks during training? Is it a fair comparison?

In section 5.2, in the online training process, if the agent interacts with the environment of task T_i to get a trajectory \tau, does the proposed method use D_i (offline dataset collected on the task T_i) to infer z and then generate reward? Or it is fine to use D_j (j is not equal to i)? How accurate is the estimated reward in comparison with the ground truth reward? Does the advantage of the proposed method heavily rely on the accuracy of the estimated reward? If yes, under the sparse-reward tasks where estimating reward is challenging (e.g. the sparse 2D navigation task in PEARL), will the proposed method fail in this case?

Is it possible to demonstrate the advantage of the proposed method on standard offline meta-RL benchmarks (e.g. D4RL)? The results on standard benchmarks with commonly used offline datasets will be more convincing.


**Summary Of The Paper:**

This work investigates the meta offline reinforcement learning problems. It highlights an issue called "distribution shift in z space" in the meta offline RL. In the meta-test phase, the agent needs to explore the test environments and the collected transitions are used to generate z representing the task feature. The policy is conditioned on the task feature z, so it is important that the transitions collected by the exploration policy are informative enough to extract the feature information. In the offline RL setting, the distritbution of transitions collected by the exploration policy deviates from the distribution of transitions in the static training dataset. Therefore, inferring task feature z becomes problematic.

This paper proposes an additional online training process (without the reward info) to fix the issue of "distribution shift in z space". It learns a reward function from the offline training dataset, and thus it is able to generate the reward for transitions collected during online training. These online training transitions with the estimated rewards are used to update the learned policy.

**Summary Of The Review:**

The online training process with generated rewards needs more clarification. And it will be better if the proposed method is evaluated on standard offline RL benchmarks.

---

> ### Author Response · Authors · 2021-11-20
> **Thank you & Response to Reviewer tmzu**
>
> Thank you for your constructive review. Below, we address the concerns brought up and answer all of your questions. In particular, below we clarify that SMAC does not have unfair access to additional training tasks and highlight that the robot manipulation task is _already_ a sparse reward task, demonstrating that SMAC performs well in this more challenging scenario. We also clarify that it is unfortunately not possible to use the D4RL benchmarks, because they are not designed as meta-learning benchmarks (they do not provide suitable multi-task distributions). Could you let us know if any other issues remain or if we misunderstood your concern?
>
> > Then when we compare with offline meta-RL (Figure 1 left) or online meta-RL (Figure 1 middle), the proposed method is exposed to more tasks during training? Is it a fair comparison?
>
> We clarify that during the self-supervised training phase, we do *not* give the SMAC agent access to new tasks. In particular, for the Half-Cheetah and Ant task, the environment is the same as in the offline data collection, but without any reward supervision. For the robot manipulation task, the agent only has access to the training tasks (without rewards) during the self-supervised phase. We show in Figure 8 in Appendix B that if we did allow the agent to train on the test tasks (without rewards, but with the held-out workspace configuration), then the performance does, unsurprisingly, improve substantially. However, for the main evaluation we did *not* allow access to these training tasks.
>
> Figure 1 is meant to highlight that additional data is collected with self-generated rewards, and not that SMAC has unfair access to new test tasks. We have updated the Figure 1 caption to clarify this.
>
> Please let us know if this resolves all of your concerns about the online training process.
>
> > Is it possible to demonstrate the advantage of the proposed method on standard offline meta-RL benchmarks (e.g. D4RL)?
>
> Thank you for the reference to D4RL. However, the D4RL benchmark is not a meta-task benchmark and there are no standard protocols for running meta-RL methods on D4RL. The D4RL benchmarks are not multi-task, and therefore require modification to adapt to this setting, so we can’t run on them directly. Due to the lack of standardized meta-RL benchmarks and datasets, we used the same tasks used in prior offline meta-RL papers [1, 2] and also created our own challenging sparse reward manipulation task to further evaluate our method. However, we believe that the tasks we propose cover the range of settings where meta-RL can be useful. We include meta-learning versions of more standard gym locomotion tasks, but these do not exhibit very high variability between tasks, and therefore are not as useful for testing generalization to new tasks (the primary use of meta-learning). We therefore also add the challenging Sawyer manipulation environments in which the agent must perform precise manipulation skills such as picking up objects and opening drawers with a diversity of objects, initial configurations, and sparse reward. In this environment, the agent sees 80 different training tasks and evaluates generalization to 10 new tasks (from the same distribution) that were not seen during training.
>
> > under the sparse-reward tasks where estimating reward is challenging (e.g. the sparse 2D navigation task in PEARL), will the proposed method fail in this case?
>
> The Sawyer Manipulation task is a sparse-reward task (0 when the desired behavior is achieved, and -1 otherwise), as discussed in paragraph 3 of Section 3. We hope that this addresses your concern about the limitation of this evaluation, as the sparse reward makes the task quite challenging, and particularly more challenging than environments used in past offline meta-RL work [1, 2].
>
> > How accurate is the estimated reward in comparison with the ground truth reward?
>
> The reward loss does not need to be particularly low for our method to work. For reference, the reward scale for the robot manipulation task is between 0 and 1. The reward decoder loss is around 0.2 on the training task, indicating that the method does not need a relatively low reward decoder loss to perform well. We have added this result to the appendix.
>
> > If yes, could the proposed unsupervised online training be beneficial to solve this issue of "distribution shift in z space" in these algorithms?
>
> Yes, as discussed in Section 4, this problem applies to any offline meta-RL and that this insight will be useful to any work on offline meta-RL [1-5]. For example, this would explain why some past work in offline meta-RL has focused on using offline datasets to perform adaptation [1], where the distribution shift between the learned exploration policy and the behavior policy does not exist. We hope that publishing this work will help practitioners and researchers be more aware of this problem.

---

> > ### Author Response · Authors · 2021-11-20
> > **Thank you & Response to Reviewer tmzu (References)**
> >
> > # References
> >
> > [1] Mitchell, Eric, et al. "Offline Meta-Reinforcement Learning with Advantage Weighting." International Conference on Machine Learning. PMLR, 2021.
> >
> > [2] Dorfman, Ron, Idan Shenfeld, and Aviv Tamar. "Offline Meta Learning of Exploration." arXiv preprint arXiv:2008.02598 (2020).
> >
> > [3] Zhao, Tony Z., et al. "Offline Meta-Reinforcement Learning for Industrial Insertion." arXiv preprint arXiv:2110.04276 (2021).
> >
> > [4] Li, Lanqing, Rui Yang, and Dijun Luo. "FOCAL: Efficient Fully-Offline Meta-Reinforcement Learning via Distance Metric Learning and Behavior Regularization." International Conference on Learning Representations. 2020.
> >
> > [5] Gupta, Priyanka, et al. "CauSeR: Causal Session-based Recommendations for Handling Popularity Bias." Proceedings of the 30th ACM International Conference on Information & Knowledge Management. 2021.

---

> ### Author Response · Authors · 2021-11-29
> **Request for Response**
>
> Dear Reviewer,
>
> We hope that you've had a chance to read our response. We would really appreciate a reply as to whether our response and clarifications have addressed the issues raised in your review, or whether there is anything else we can address.
>
> Thank you.

---

### Official Review · Reviewer_w6Yf · 2021-11-02

**Correctness:** 4
**Technical Novelty And Significance:** 3
**Empirical Novelty And Significance:** 4
**Recommendation:** 6
**Confidence:** 4

**Main Review:**

## Paper strengths and contributions
**Motivation and intuition**
- Tackling distributional shifts of the exploration policy in offline meta-RL is well motivated.
- Introducing the online self-supervised learning phase is intuitive and convincing.

**Novelty**
- I believe the introduced online self-supervised learning phase is novel.
- The proposed self-supervised learning method is an intuitive way to learn from the online self-supervised learning phase.

**Clarity**
- Writing is easy to understand. Approach, notations, and figures are well explained.
- Figure 2 well supports the targeting problem and motivation.
- Figure 6 qualitatively showed targeting problem is resolved.

**Related work**
- Authors clearly address the additional assumption (online phase) from related work and its motivation.

**Experimental results**
- The presentation of the experimental results is clear.
- The experimental results demonstrate that the proposed approach
Figure 5 shows that self-supervised learning gradually improves meta-agent even without reward labels.

**Reproducibility**
- The code is provided, which helps understand the details of the proposed framework.
- Appendix gives essential details for implementation. I believe reproducing the results is possible.

## Paper weaknesses and questions

**Offline datasets**
This work (and offline meta-RL prior work) assume the availability of offline/static datasets labeled with rewards for different tasks, which could be very expensive to obtain in some cases, limiting the applicability of this line of work. This work uses scripted policies to obtain the offline datasets, which means that all the tasks are solvable already. I would like to hear the authors' opinions on how this can scale up to more complex domains.

**Suboptimal data**
The robot succeeds on the task in 46% of the transitions in the offline dataset, which is a pretty low success rate. I wonder if the proposed framework works better partially because of the suboptimality of this dataset.

**Online interactions**
The proposed method needs additional online interactions compared to offline meta-RL prior works. While it could be cheaper to collect, sometimes it can still be impossible to leverage online interactions.

**Reward prediction performance**
It would be informative to see how well it can predict the rewards of online interactions and how this affects learning performance.

## Other metrics

### Relevance and significance
Solid contributions to a relevant problem

### Novelty
Worthy contributions, but not surprising

### Technical quality
Technically adequate for its area, solid results

### Experimental evaluation
Solid, informative evaluation w.r.t all 5 criteria

### Clarity
Very clear, only minor flaws.

**Summary Of The Paper:**

Meta-RL methods aim to learn to learn using a set of meta-training tasks. Yet, it requires a large number of online interactions for training. Offline meta-RL methods aim to learn from static datasets labeled with rewards for different tasks. These methods usually suffer from distributional shifts between the behavior policy from the offline data and the meta-test time exploration policy. This paper aims to tackle this distributional shift issue of offline meta-RL methods. To this end, the paper proposes a framework that is robust to the distributional shift by introducing a self-supervised learning phase. In this phase, the proposed framework leverages online data without reward labels and generates synthetic reward labels for it based on the labeled offline data, allowing for smoother adaptation. The experiments show the proposed approach alleviates the distribution shift issue and thus marginally improves post-adaptation results compared to offline meta-RL prior works. I believe this work tackles an interesting and promising research direction (i.e. alleviate the distributional shift issue in offline meta-RL) and proposes a convincing framework to address it. The experimental results verify the effectiveness of the proposed framework. Therefore, I am leaning toward accepting this paper with some concerns on the assumptions made by this work and its applicability.

**Summary Of The Review:**

I believe this work tackles an interesting and promising research direction (i.e. alleviate the distributional shift issue in offline meta-RL) and proposes a convincing framework to address it. The experimental results verify the effectiveness of the proposed framework. Therefore, I am leaning toward accepting this paper with some concerns on the assumptions made by this work and its applicability.

---

> ### Author Response · Authors · 2021-11-22
> **Thank you & Response to Reviewer w6Yf**
>
> Thank you for your constructive review. Below, we address the comment regarding how feasible it is to obtain an offline dataset (of note, our method has already been applied to real-world applications), discuss the use of suboptimal data, and provide evidence that our method does not require high reward prediction performance.
>
> We believe we have addressed all of your concerns, but please let us know if we have missed anything.
>
> > This work (and offline meta-RL prior work) assume the availability of offline/static datasets labeled with rewards for different tasks, which could be very expensive to obtain in some cases, limiting the applicability of this line of work
>
> We acknowledge that the availability of offline data is an important assumption to our work, and that this assumption may not always be satisfied. However, we note that there are many applications in which prior data for RL is available, such as in these works [1, 2, 3, 4]. In fact, **researchers have successfully scaled up our method to real-world robot applications**, as shown in this anonymized link:
> https://drive.google.com/file/d/1E9v5OQgYCPf6lRfjdvtouKT8RFSYVvR-/view?usp=sharing
>
> (Note: we have confirmed with the AC that we have sufficiently anonymized this link and redacted the PDF and that it is okay for us to share.) Lastly, many prior works study this offline RL setting (albeit in non-meta learning settings), further evidence that the community considers the assumption to be reasonable [5, 6, 7, 8, 9, 10].
>
> > Suboptimal data... I wonder if the proposed framework works better partially because of the suboptimality of this dataset.
>
> We agree with your intuition. An assumption is that the offline data is not optimal. If the offline data is already optimal, then it is quite possible that offline RL is unnecessary, because behavior cloning is quite competitive since there is little room for improvement [11]. Our experiments were designed to test if SMAC can improve over a simple (meta) behavior cloning when the data is sub-optimal, which we found was the case (see Figure 5). We focused on the scenario where offline data is suboptimal because this is a realistic scenario and of interest to the community [5]. We have added an abbreviation explanation to Section 6.
>
> > Reward prediction performance It would be informative to see how well it can predict the rewards of online interactions and how this affects learning performance.
>
> Thank you for this suggestion. To address this concern, we have added the following result to the appendix, which shows that **the reward loss does not need to be particularly low for our method to work**: As an example, the maximum reward magnitude for the robot manipulation task is 1. The reward decoder loss is around 0.2 on the training task, indicating that the method does not need a relatively low reward decoder loss to perform well.
>
> ### References
>
> [1] ​​Lewis, Mike, et al. "Deal or No Deal? End-to-End Learning of Negotiation Dialogues." Proceedings of the 2017 Conference on Empirical Methods in Natural Language Processing. 2017.
>
> [2] Kalashnikov, Dmitry, et al. "Qt-opt: Scalable deep reinforcement learning for vision-based robotic manipulation." arXiv preprint arXiv:1806.10293 (2018).
>
> [3] Cabi, Serkan, et al. "Scaling data-driven robotics with reward sketching and batch reinforcement learning." arXiv preprint arXiv:1909.12200 (2019).
>
> [4] Suo, Daniel, et al. “Machine Learning for Mechanical Ventilation Control.” arXiv preprint arXiv:2102.06779 (2021).
>
> [5] Fujimoto, Scott, David Meger, and Doina Precup. "Off-policy deep reinforcement learning without exploration." International Conference on Machine Learning. PMLR, 2019.
>
> [6] Wu, Yifan, George Tucker, and Ofir Nachum. "Behavior regularized offline reinforcement learning." arXiv preprint arXiv:1911.11361 (2019).
>
> [7] Nadjahi, Kimia, Romain Laroche, and Rémi Tachet des Combes. "Safe policy improvement with soft baseline bootstrapping." arXiv preprint arXiv:1907.05079 (2019).
>
> [8] Peng, Xue Bin, et al. "Advantage-weighted regression: Simple and scalable off-policy reinforcement learning." arXiv preprint arXiv:1910.00177 (2019).
>
> [9] Kumar, Aviral, et al. "Conservative q-learning for offline reinforcement learning." arXiv preprint arXiv:2006.04779 (2020).
>
> [10] Singh, Avi, et al. "Cog: Connecting new skills to past experience with offline reinforcement learning." arXiv preprint arXiv:2010.14500 (2020).
>
> [11] Spencer, Jonathan, et al. "Feedback in Imitation Learning: The Three Regimes of Covariate Shift." arXiv preprint arXiv:2102.02872 (2021).

---

> > ### Comment · Reviewer_w6Yf · 2021-11-23
> > **Re: Thank you & Response to Reviewer w6Yf**
> >
> > Thank you addressing some of my concerns.  I have decided to keep my original rating and am slightly leaning toward accepting this paper.

---

### Official Review · Reviewer_RpMh · 2021-11-02

**Correctness:** 3
**Technical Novelty And Significance:** 2
**Empirical Novelty And Significance:** 3
**Recommendation:** 3
**Confidence:** 4

**Main Review:**

## Merits

1.	The problem this paper aims to address is practically important for RL, that is, how to effectively utilize offline datasets to meta-train policies and enable them for fast adaptation to new tasks.
2.	This paper identifies a specific problem to offline Meta-RL : distribution shift of the exploration policy for collecting data for adaption, which looks interesting.
3.	The idea of using unsupervised online data collection seems novel and shows some effectiveness in alleviating the problem of the distribution shift during the adaptation.
4.	This paper is generally well-written and easy to follow.


 ## Concerns and limitations
1.	The technical contribution of the proposed approach is limited. It is basically a simple combination of AWAC and PEARL. One modification to this combination is to use unsupervised online data collection, whose data are labeled with synthetic rewards, instead of ground-true rewards. However, synthetic reward learning is not novel and looks straightforward.
2.	Another concern is that the proposed approach seems to assume tasks share the same dynamic model and only differ in reward functions. This is because the proposed SMAC method learns the task encoder only through the backpropagation from the reward learning, in contrast to prior work like PEARL. This assumption may bring SMAC with some advantages over other more general methods in tasks only differing with rewards, but will significantly limit its applicability. How will SMAC perform in settings where tasks may have different dynamics?
3.	The reviewer is also curious about how this approach works in sparse-reward tasks?
4.	From the experiments, the outperformance of SMAC is mainly due to the unsupervised online data collection, which may not be a fair comparison to baselines. Without this online data collection, SMAC underperforms MACAW in two out of three tasks.
5.	This paper only evaluates the proposed method in three scenarios. It is highly recommended to include more settings to strengthen the empirical results.
6.	In Algorithm 1, Line 8 is confusing. What is $D_i$ in this line?



**Summary Of The Paper:**

This paper aims to enable effective meta-RL in the offline setting. It first identifies the problem of simply combining meta-RL with offline RL: distribution shift in z-space. This paper then proposes a hybrid method that uses offline datasets to meta-train an adaptive policy and then collects and utilizes additional unsupervised online data to bridge the identified distribution shift. Empirical results show that unsupervised online learning significantly improves the adaptive capabilities of the meta-trained policy.

**Summary Of The Review:**

This paper identifies a specific problem of distribution drift in offline Meta-RL and proposes an interesting idea of using unsupervised data collection to alleviate this problem. This paper is generally well-written. However, the proposed approach has limited technical contributions and makes strong implicit assumptions. The experimental evaluation also needs to be improved by including more settings.

---

> ### Author Response · Authors · 2021-11-20
> **Thank you & Response to Reviewer RpMh (1/2)**
>
> Thank you for your constructive review. Below, we address your concerns regarding the technical contribution, assumptions, and experimental evaluation. In particular, we have added additional experiments on three challenging tasks, including tasks with changing dynamics. We also note that our newly proposed robot manipulation task shown in Figure 4 is **already** a sparse-reward task and is significantly more diverse than domains used in past work. Lastly, we highlight that our technical contribution is not just the particular method, but in fact a combination of analysis of distributional shift, the technical approach, and the addition of unlabeled data.
>
> Could you let us know if you have any remaining concerns or if we misunderstood your concerns?
>
> # Additional Experiments
> > This paper only evaluates the proposed method in three scenarios. It is highly recommended to include more settings to strengthen the empirical results.
>
> As suggested, we have run additional experiments on three challenging environments: Walker, Hopper, and Humanoid. These environments are challenging due to their high dimensional state and action space (e.g. the Humanoid state space has 376 dimensions). Moreover, in the Walker and Hopper environments, rather than changing the rewards, we vary the dynamics by changing some of the environment parameters (e.g. mass), as done in past meta-RL work [1,27,28]. We see in Figure 5 of the updated revision that our method performs well and significantly outperforms the naive meta-BC baseline. We're working on adding the performance of MACAW and BOReL and will update the plot over the weekend.
>
> > Another concern is that the proposed approach seems to assume tasks share the same dynamic model and only differ in reward functions...How will SMAC perform in settings where tasks may have different dynamics?
>
> We hope that our additional experiments address this concern. We note that although the decoder is trained to only predict the reward, we found that our method still works on these tasks that require genearlize to new dynamics. We hypothesize that this is because the rewards implicitly depend on the dynamics (since the rewards depend on the velocity).
>
> We also highlight that the tasks in the Sawyer Manipulation differ much more than in only their reward function. The configuration of the initial state changes significantly between episodes (as shown Figure 4), and we believe that this is reflective of practical robot applications. For example, although the laws of physics are constant, a robot in the real world must constantly adapt to new settings.
>
> One could change our method to backpropagate the loss from the Q-function, which would encourage the latent space to capture information about the dynamics, but we did not find this to be necessary in our experiments.
>
> # Technical Contribution
> > The technical contribution of the proposed approach is limited. It is basically a simple combination of AWAC and PEARL. One modification to this combination is to use unsupervised online data collection, whose data are labeled with synthetic rewards, instead of ground-true rewards. However, synthetic reward learning is not novel and looks straightforward.
>
> An important technical contribution of this work is highlighting and addressing the additional distribution shift that occurs with offline meta-RL, which we believe is a hitherto unidentified phenomena in offline meta-RL. Given the abundance of work in this field [1,2,3,4,5] we believe that this new insight, as well as the accompanying empirical analysis, is of great interest to the research community.
>
> Although our method may be simple to describe, we do not see this as a technical limitation nor has this been the standard for accepting papers. On the contrary, many significant papers in the field are significant because of their contribution to the community’s understanding, despite having a fix that is relatively simple to implement (e.g. taking the min between two functions [6], adding a regularization loss [7], or clipping certain values [8]).
>
> If the reviewer is aware of past work that has highlighted, studied, and addressed the issue of distribution shift in offline meta-RL, we would be happy to adjust the claims in our paper appropriately.

---

> > ### Author Response · Authors · 2021-11-20
> > **Thank you & Response to Reviewer RpMh (2/2)**
> >
> > # Experimental Evaluation
> > > The reviewer is also curious about how this approach works in sparse-reward tasks?
> >
> > The Sawyer Manipulation task **is a already sparse-reward task** (0 when the desired behavior is achieved, and -1 otherwise), as discussed in paragraph 3 of Section 3. We hope that this addresses your concern about the limitation of this evaluation, as the sparse reward makes the task quite challenging, and particularly more challenging than environments used in past offline meta-RL work [1, 2].
> > > From the experiments, the outperformance of SMAC is mainly due to the unsupervised online data collection, which may not be a fair comparison to baselines.
> >
> > Since we study a different problem setting from past work, we cannot compare directly to these past work. In particular, MACAW and BOReL assume that rewards are provided during training and cannot make use of the self-supervised online interaction.
> >
> > We note that our primary contribution--that of using self-supervised interaction to mitigate distribution shift offline meta RL--is largely orthogonal to the underlying meta RL algorithm and could be combined with MACAW and BOReL as well. In this paper, we built off of PEARL due to its simplicity, and believe that our insight will be used to improve other offline meta-RL papers in the future.
> >
> > We have added this clarification to Section 6 and the Conclusion.
> >
> > > Without this online data collection, SMAC underperforms MACAW in two out of three tasks.
> >
> > We acknowledge that SMAC does not consistently outperform MACAW before the self-supervised phase. However, we also believe that there are a number of settings in which data collection is cheap and researchers and practitioners are interested in performance _after_ a self-supervised phase [9, 10, 11]. For example, a popular paradigm is to train robot agents in simulation and later deploy them in the real world with sim-to-real transfer techniques [12-15]. In these domains, data collection is cheap. Outside of robotics, training web-based [16], video game [17], and language agents [18-21] can also benefit from relatively cheap, self-supervised environment interactions. Lastly, we note that there is concerted effort in developing robot platforms that already support safe and automatic reset [22-26]. In summary, we agree that SMAC does not always outperform MACAW before the self-supervised phase, but believe that the strong performance after self-supervised learning would be of interest to a number of other researchers and practitioners.
> >
> > ## Miscellaneous
> > > 6. In Algorithm 1, Line 8 is confusing. What is $$D_i$$ in this line?
> >
> > Thank you for pointing out this omission. The buffer $$D_i$$ is the same buffer used to generate the reward, as discussed in Section 5.2. We have added this clarification to Algorithm 1.

---

> > > ### Author Response · Authors · 2021-11-20
> > > **Thank you & Response to Reviewer RpMh (References)**
> > >
> > > # References
> > > [1] Mitchell, Eric, et al. "Offline Meta-Reinforcement Learning with Advantage Weighting." International Conference on Machine Learning. PMLR, 2021.
> > >
> > > [2] Dorfman, Ron, Idan Shenfeld, and Aviv Tamar. "Offline Meta Learning of Exploration." arXiv preprint arXiv:2008.02598 (2020).
> > >
> > > [3] Zhao, Tony Z., et al. "Offline Meta-Reinforcement Learning for Industrial Insertion." arXiv preprint arXiv:2110.04276 (2021).
> > >
> > > [4] Li, Lanqing, Rui Yang, and Dijun Luo. "FOCAL: Efficient Fully-Offline Meta-Reinforcement Learning via Distance Metric Learning and Behavior Regularization." International Conference on Learning Representations. 2020.
> > >
> > > [5] Gupta, Priyanka, et al. "CauSeR: Causal Session-based Recommendations for Handling Popularity Bias." Proceedings of the 30th ACM International Conference on Information & Knowledge Management. 2021.
> > >
> > > [6] Fujimoto, Scott, Herke Hoof, and David Meger. "Addressing function approximation error in actor-critic methods." International Conference on Machine Learning. PMLR, 2018.
> > >
> > > [7] Kumar, Aviral, et al. "Conservative q-learning for offline reinforcement learning." arXiv preprint arXiv:2006.04779 (2020).
> > >
> > > [8] Schulman, John, et al. "Proximal policy optimization algorithms." arXiv preprint arXiv:1707.06347 (2017).
> > >
> > > [9] Colas, Cédric, et al. "CURIOUS: intrinsically motivated modular multi-goal reinforcement learning." International conference on machine learning. PMLR, 2019.
> > >
> > > [10] Pong, Vitchyr, et al. "Skew-Fit: State-Covering Self-Supervised Reinforcement Learning." International Conference on Machine Learning. PMLR, 2020.
> > >
> > > [11] Sekar, Ramanan, et al. "Planning to explore via self-supervised world models." International Conference on Machine Learning. PMLR, 2020.
> > >
> > > [12] Peng, Xue Bin, et al. "Sim-to-real transfer of robotic control with dynamics randomization." 2018 IEEE international conference on robotics and automation (ICRA). IEEE, 2018.
> > >
> > > [13] Y. Chebotar et al., "Closing the Sim-to-Real Loop: Adapting Simulation Randomization with Real World Experience," 2019 International Conference on Robotics and Automation (ICRA), 2019, pp. 8973-8979, doi: 10.1109/ICRA.2019.8793789.
> > >
> > > [14] Hwangbo, Jemin, et al. "Learning agile and dynamic motor skills for legged robots." Science Robotics 4.26 (2019).
> > >
> > > [15] Andrychowicz, OpenAI: Marcin, et al. "Learning dexterous in-hand manipulation." The International Journal of Robotics Research 39.1 (2020): 3-20.
> > >
> > > [16] Torrado, Ruben Rodriguez, et al. "Deep reinforcement learning for general video game ai." 2018 IEEE Conference on Computational Intelligence and Games (CIG). IEEE, 2018.
> > >
> > > [17] Nogueira, Rodrigo, and Kyunghyun Cho. "End-to-end goal-driven web navigation." Advances in neural information processing systems 29 (2016): 1903-1911.
> > >
> > > [18] Zhong, Victor, Caiming Xiong, and Richard Socher. "Seq2sql: Generating structured queries from natural language using reinforcement learning." arXiv preprint arXiv:1709.00103 (2017).
> > >
> > > [19] Gupta, Rahul, Aditya Kanade, and Shirish Shevade. "Deep reinforcement learning for programming language correction." arXiv preprint arXiv:1801.10467 (2018).
> > >
> > > [20] He, Ji, et al. "Deep Reinforcement Learning with a Natural Language Action Space." Proceedings of the 54th Annual Meeting of the Association for Computational Linguistics (Volume 1: Long Papers). 2016.
> > >
> > > [21] He, Dongliang, et al. "Read, watch, and move: Reinforcement learning for temporally grounding natural language descriptions in videos." Proceedings of the AAAI Conference on Artificial Intelligence. Vol. 33. No. 01. 2019.
> > >
> > > [22] Levine, Sergey, et al. "Learning hand-eye coordination for robotic grasping with deep learning and large-scale data collection." The International Journal of Robotics Research 37.4-5 (2018): 421-436.
> > >
> > > [23] Yang, Brian, et al. "Replab: A reproducible low-cost arm benchmark platform for robotic learning." International Conference on Robotics and Automation. 2019.
> > >
> > > Neunert, Michael, et al. "Continuous-discrete reinforcement learning for hybrid control in robotics." Conference on Robot Learning. PMLR, 2020.
> > >
> > > [24] Nagabandi, Anusha, et al. "Learning to Adapt in Dynamic, Real-World Environments through Meta-Reinforcement Learning." International Conference on Learning Representations. 2018.
> > >
> > > [25] Bloesch, Michael, et al. "Towards Real Robot Learning in the Wild: A Case Study in Bipedal Locomotion." 5th Annual Conference on Robot Learning. 2021.
> > >
> > > [26] Gupta, Abhishek, et al. "Reset-Free Reinforcement Learning via Multi-Task Learning: Learning Dexterous Manipulation Behaviors without Human Intervention." International Conference on Robotics and Automation. 2021.
> > >
> > > [27] Rakelly, Kate, et al. "Efficient off-policy meta-reinforcement learning via probabilistic context variables." International conference on machine learning. PMLR, 2019.
> > >
> > > [28] Rothfuss, Jonas, et al. "ProMP: Proximal Meta-Policy Search." International Conference on Learning Representations. 2018.

---

> > > > ### Author Response · Authors · 2021-11-22
> > > > **New MACAW results added on Hopper Param, Walker Param, and Humanoid Direction.**
> > > >
> > > > We have added the results from running MACAW on Hopper Param, Walker Param, and Humanoid Direction to Figure 5. As before, we used the author’s code for running the MACAW baselines. For all of these new experiments, we did not tune any of the methods and instead used the same hyperparameters as from the Ant task. This experiment additionally tests the sensitivity of the different methods to the environment. The results demonstrate that SMAC consistently outperforms this prior work and does not require environment-specific hyperparameters.
> > > >
> > > > We tried to run BOReL but could not get it to work on the new environments. However, we will make our best effort to include the BOReL results for the final.

---

> > > > > ### Author Response · Authors · 2021-11-29
> > > > > **Request for Response**
> > > > >
> > > > > Dear Reviewer,
> > > > >
> > > > > We hope that you've had a chance to read our response. We would really appreciate a reply as to whether our response and clarifications have addressed the issues raised in your review, or whether there is anything else we can address.
> > > > >
> > > > > Thank you.

---

### Official Review · Reviewer_fffq · 2021-11-04

**Correctness:** 3
**Technical Novelty And Significance:** 3
**Empirical Novelty And Significance:** 2
**Recommendation:** 6
**Confidence:** 3

**Main Review:**

Strengths
+ It is a simple addition to the meta-RL setup and is shown to alleviate the z-distribution shift.
+ Related work is well covered.

Weakness/Comments
- Some of the exposition is a bit high-level. Grounding in running examples would be helpful to better understand the overall approach. For example, in a set of tasks how different can any two tasks, is the difference at the level of one being picking and one being opening a drawer or is the difference at the level of different pick locations in a pick task.
- '... can autonomously interact with the environment ...' is a very strong assumption for many real world tasks. This needs automatic resets in place as well as safe policies that already allow the agent to explore (which if are already available and robust defeats the purpose of learning to some extent?).
- The claim that '... reward prediction generalization is easier than policy state, action, and z generalization' is not sufficiently supported. If the reward decoder is not well trained or reliable then wouldn't the bad labels skew the learning in the wrong direction? It wasn't clear how this issue was being addressed or if this imposes limits to the overall approach and where it can be used.

**Summary Of The Paper:**

This paper introduces a reward prediction module to produce reward labels in a meta-RL setting. Here the offline RL part requires reward labels while the online portion can be done in a self-supervised manner by generating reward labels using the offline trained module.


**Summary Of The Review:**

The idea is simple and works on the problems tested. The larger applicability and constraints of the approach are not clear. Explanation of the approach can be improved.

---

> ### Author Response · Authors · 2021-11-20
> **Thank you & Response to Reviewer fffq**
>
> Thank you for your helpful feedback. We have updated our paper to include your suggestions, clarify what we meant by “reward prediction generalization is easier” and the evidence for this, and discuss the prevalence of our problem setting, as we detail below.
>
> > The claim that '... reward prediction generalization is easier than policy state, action, and z generalization' is not sufficiently supported. If the reward decoder is not well trained or reliable then wouldn't the bad labels skew the learning in the wrong direction?
>
> We clarify that we _hypothesize_ that reward prediction generalization is easier than policy state, action, and $z$ generalization. If true, then we would expect that using the reward decoder to label rewards and train on those labels, as in SMAC, will outperform directly using the policy on new tasks (as in Figure 2). We see that this is the case, as shown in Figure 6. We have added this clarification to the paper.
>
> > Some of the exposition is a bit high-level. Grounding in running examples would be helpful to better understand the overall approach
>
> Thank you for the suggestion. We have added a robot manipulation example to Section 4 when explaining the distribution shift. Specifically, we discuss how an offline dataset may contain smooth trajectories that were collected by a human teleoperator. In contrast, the learned exploration may contain jittering due learning artifacts or the use of a stochastic policy. This jittering may not impede the robot from exploring the environment, but may result in a trajectory distribution shift that degrades the adaptation process, which only learned to adapt to smooth, human-generated trajectories in the offline dataset.
>
> > This needs automatic resets in place as well as safe policies
>
> We agree that the ability to reset the environment to perform self-supervised exploration is an important assumption. This assumption may not be satisfied due to safety concerns and engineering needed to ensure that automatic resets are feasible, and therefore this is an important direction for future research. We have updated the conclusion to acknowledge this limitation. However, we still believe that the assumption is satisfied in a number of domains and settings that may interest certain researchers and practitioners in the community. For example, a popular paradigm is to train robot agents in simulation and later deploy them in the real world with sim-to-real transfer techniques [1, 2, 3, 4]. In these domains, automatic and safe resets are trivial. Outside of robotics, these techniques may be useful in training web-based agents [5], video game agents [6], and language agents [7, 8, 9, 10], in which resetting is effectively free. Lastly, we note that there is concerted effort in developing robot platforms that already support safe and automatic reset [11, 12, 13, 14, 15]. In summary, we agree that this is an important assumption, which is not always satisfied, but is satisfied in a meaningful number of settings.

---

> > ### Author Response · Authors · 2021-11-20
> > **Thank you & Response to Reviewer fffq (References)**
> >
> > ## References
> >
> > [1] Peng, Xue Bin, et al. "Sim-to-real transfer of robotic control with dynamics randomization." 2018 IEEE international conference on robotics and automation (ICRA). IEEE, 2018.
> >
> > [2] Y. Chebotar et al., "Closing the Sim-to-Real Loop: Adapting Simulation Randomization with Real World Experience," 2019 International Conference on Robotics and Automation (ICRA), 2019, pp. 8973-8979, doi: 10.1109/ICRA.2019.8793789.
> >
> > [3] Hwangbo, Jemin, et al. "Learning agile and dynamic motor skills for legged robots." Science Robotics 4.26 (2019).
> >
> > [4] Andrychowicz, OpenAI: Marcin, et al. "Learning dexterous in-hand manipulation." The International Journal of Robotics Research 39.1 (2020): 3-20.
> >
> > [5] Torrado, Ruben Rodriguez, et al. "Deep reinforcement learning for general video game ai." 2018 IEEE Conference on Computational Intelligence and Games (CIG). IEEE, 2018.
> >
> > [6] Nogueira, Rodrigo, and Kyunghyun Cho. "End-to-end goal-driven web navigation." Advances in neural information processing systems 29 (2016): 1903-1911.
> >
> > [7] Zhong, Victor, Caiming Xiong, and Richard Socher. "Seq2sql: Generating structured queries from natural language using reinforcement learning." arXiv preprint arXiv:1709.00103 (2017).
> >
> > [8] Gupta, Rahul, Aditya Kanade, and Shirish Shevade. "Deep reinforcement learning for programming language correction." arXiv preprint arXiv:1801.10467 (2018).
> >
> > [9] He, Ji, et al. "Deep Reinforcement Learning with a Natural Language Action Space." Proceedings of the 54th Annual Meeting of the Association for Computational Linguistics (Volume 1: Long Papers). 2016.
> >
> > [10] He, Dongliang, et al. "Read, watch, and move: Reinforcement learning for temporally grounding natural language descriptions in videos." Proceedings of the AAAI Conference on Artificial Intelligence. Vol. 33. No. 01. 2019.
> >
> > [11] Levine, Sergey, et al. "Learning hand-eye coordination for robotic grasping with deep learning and large-scale data collection." The International Journal of Robotics Research 37.4-5 (2018): 421-436.
> >
> > [12] Yang, Brian, et al. "Replab: A reproducible low-cost arm benchmark platform for robotic learning." International Conference on Robotics and Automation. 2019.
> > Neunert, Michael, et al. "Continuous-discrete reinforcement learning for hybrid control in robotics." Conference on Robot Learning. PMLR, 2020.
> >
> > [13] Nagabandi, Anusha, et al. "Learning to Adapt in Dynamic, Real-World Environments through Meta-Reinforcement Learning." International Conference on Learning Representations. 2018.
> >
> > [14] Bloesch, Michael, et al. "Towards Real Robot Learning in the Wild: A Case Study in Bipedal Locomotion." 5th Annual Conference on Robot Learning. 2021.
> >
> > [15] Gupta, Abhishek, et al. "Reset-Free Reinforcement Learning via Multi-Task Learning: Learning Dexterous Manipulation Behaviors without Human Intervention." International Conference on Robotics and Automation. 2021.

---

> > > ### Comment · Reviewer_fffq · 2021-11-29
> > > **Thank you for the response**
> > >
> > > Thanks for answering the questions and acknowledge the constraint with resets but also pointing out the broader context in which this is still okay. I will keep my original score.

---

### Decision · Program_Chairs · 2022-01-20

**Decision:**

Reject

**Comment:**

The paper describes a new offline meta RL technique that addresses the distributional shift problem with a self-supervised online exploration phase where reward labels are not available.  The framework is novel and interesting.  The authors addressed many concerns of the reviewers.  However, the additional experiments raised additional questions.  For instance, why does meta-BC perform so well, even better than the proposed method without online data, and other baselines seem not to work at all?  In the discussion, the reviewers expressed concerns about the experimental results in the case of changing dynamics.  Those experiments are questionable since the proposed method only considers the reward information to deal with different dynamics.  Finally, an important question regarding SMAC remains unanswered: how much does the proposed method depend on the quality of the offline dataset and the quality of the reward decoder? Overall, the work is promising and the authors are encouraged to continue their work by addressing the reviewers concerns.